# Multiple exciton generation boosting over 100% quantum efficiency photoelectrochemical photodetection

Junjun Xue[1], Xu Wang[1], Guanyu Xu[1], Xinya Tao[1], Tongdao Pan[1], Zhouyu Chen[2], Qing Cai [ID][3], Pengfei Shao[3] ✉, Guofeng Yang[4] ✉, Zengli Huang[5], Ting Zhi[1], Ke Wang[3], Bin Liu [ID][3], Dunjun Chen[3] ✉, Rong Zhang [ID][3] & Jin Wang [ID][1] ✉

The self-powered photoelectrochemical components themselves featured advancements in operating independently without external supply. Ultimately, due to lack of assistance from the external bias, the photoelectrochemical response is commonly restricted by the deficient photo-quantum efficiency for the absence of carrier multiplication. This work demonstrates a self-powered photoelectrochemical photodetector based on $CuO_x$/AlGaN nanowires with staggered band structure and enhanced built-in potential for efficient exciton extraction. The generated multiple excitons within reach-through $CuO_x$ layer could be speedily separated before Auger recombination. This yields a 131.5% external quantum efficiency and 270.6 mA W$^{-1}$ responsivity at 255 nm. The work confirms the role of multiple exciton generation in photoelectrochemical systems, offering a solution on paving path of advance for self-powered optoelectronics and weak-light UV imaging applications.

Self-powered electronics based on photic driving can operate independently and sustainably, which are considered as core components in the low-carbon industrial system. Self-powered photodetectors (PDs), for instance, are indispensable modules for constructing energy-saving optoelectronic systems[1,2]. Among the existing self-powered PDs, the developing photoelectrochemical (PEC) PDs stand out due to their low cost, non-lithographic fabrication, robust water-resistance and multifunctional adjustable light response, making it receive dramatic increasing attentions in recent years[3,4]. However, for these conventional self-powered PEC PDs, the photoelectric response cannot benefit from the internal photoelectric gain aroused by the assistance of a power supply and thus the corresponding responsivity is suppressed by the deficient quantum efficiency[5] (normally not exceeding 100%).

Generally, acquiring access to the photoelectric gain effects in solid-state photodetectors, such as avalanche photodiodes[6], photomultipliers[7], and photodetectors combined with low-noise trans-resistance amplifiers[8], is an effective method to achieve the ultra-high quantum efficiency. However, all the cases above are inevitable to require for the external supplied voltage. In fact, introducing the multiple exciton generation (MEG) effect into the self-powered PEC PDs is a feasible approach to beat the gain-voltage tradeoff. In 2001, Nozik predicted that the highly efficient multiple exciton generation effect in nano-semiconductors[9,10]. Due to quantum size effects, the bandgap ($E_g$) of nano-semiconductors can be effectively tuned by adjusting their size[11], leading to the formation of discrete intra-band energy levels[12]. The discreteness of energy reduces the probability of phonon-mediated relaxation, slowing down the electron relaxation process and effectively suppressing the Auger recombination effect. In addition, the reduction in nano-semiconductor size increases the surface-to-volume ratio, resulting in a higher density of surface electrons, which intensifies the Coulomb interaction[11]. Upon absorbing a high-energy photon, a hot electron is created in the nano-semiconductor. Due to the effective suppression over Auger

[1]GaN-X Laboratory, College of Electronic and Optical Engineering & College of Flexible Electronics (Future Technology), Nanjing University of Posts and Telecommunications, Nanjing, China. [2]Portland Institute of NJUPT, Nanjing University of Posts and Telecommunications, Nanjing, China. [3]Key Laboratory of Advanced Photonic and Electronic Materials, School of Electronic Science and Engineering, Nanjing University, Nanjing, China. [4]School of Science, Jiangnan University, Wuxi, China. [5]Suzhou Laboratory, Suzhou, China. ✉e-mail: pfshao@nju.edu.cn; gfyang@jiangnan.edu.cn; djchen@nju.edu.cn; jin@njupt.edu.cn

recombination and the enhancement on Coulomb interaction, the hot electron with sufficient kinetic energy could avoid cooling via radiating phonons but instead excites other ground-state electrons to higher energy states through collisions[13]. In this process, if the photon energy ($E$) is above the energy threshold ($E_{th} > 2E_g$), a high-energy photon can generate two or more electron-hole pairs in the nano-semiconductor[14–16], resulting in internal photoelectric gain and breaking through the restriction of the quantum efficiency.

To achieve the effect of photoelectric gain without bias, aside from the MEG by high-energy photon excitation, an appropriate band alignment and the internal electric field intensity enhancement should be accounted in, which facilitates exciton extraction and carrier transfer over the thermal exciton relaxation process[17–19]. Herein, a PEC PD based on the p-type CuO·Cu$_2$O (CuO$_x$) nanocomposite/n-type AlGaN nanowire heterostructure is developed to achieve high photoelectric conversion efficiency. Under 255 nm UV light illumination and 0 V bias, the high-energy electrons in the CuO$_x$ auxiliary light-absorbing layers on AlGaN NWs are excited from the valence band to an energy level above the minimum of the conduction band, resulting in collision ionization and an efficient MEG process. The strong built-in electric field at the AlGaN/CuO$_x$ type-II heterojunction interface promotes the separation of multiple carriers without cooling, significantly enhancing the photocurrent. The AlGaN/CuO$_x$ UV PD exhibits a high responsivity of 270.6 mA W$^{-1}$ and an external quantum efficiency (EQE) of 131.5% at the light intensity of 5 μW cm$^{-2}$, achieving one of the highest results among the reported self-powered PEC PDs to date. Due to excellent photoresponse, the efficient AlGaN/CuO$_x$ PEC PD can be applied for high-quality UV imaging techniques. This work provides a method for developing the next generation of high-performance PEC UV detectors.

## Results

### Reach-through band bending at the solid−electrolyte interface

The photoelectrode, adopted for this work, is composed of Si-doped n-Al$_{0.3}$Ga$_{0.7}$N nanowires (~400 nm long), vertically grown on n-Si (111) substrate, as shown in Fig. 1a and Supplementary Fig. 1. The AlGaN-based PEC PDs were fabricated using an encapsulation technique to access the PEC response of AlGaN-based photoelectrodes (see "Methods"). A typical three-electrode quartz window cell was employed (Fig. 1a), consisting of a working electrode (PEC PD), a counter electrode (Pt), a reference electrode (Ag/AgCl), and the electrolyte solution (Na$_2$SO$_4$). When bare n-type AlGaN contacts the electrolyte, upward band bending occurs on the surface of AlGaN, resulting in positive photocurrent under deep-UV light irradiation (detailed mechanism of carrier transport can be seen in Supplementary Fig. 2a).

In order to modify the surface physicochemical properties and control the behavior of carriers at the surface, the rational metal oxides decoration on nanostructural semiconductor is an effective strategy for the pursuit of PEC devices with superior performance. Herein, the n-AlGaN nanowires were treated with surface modulation by

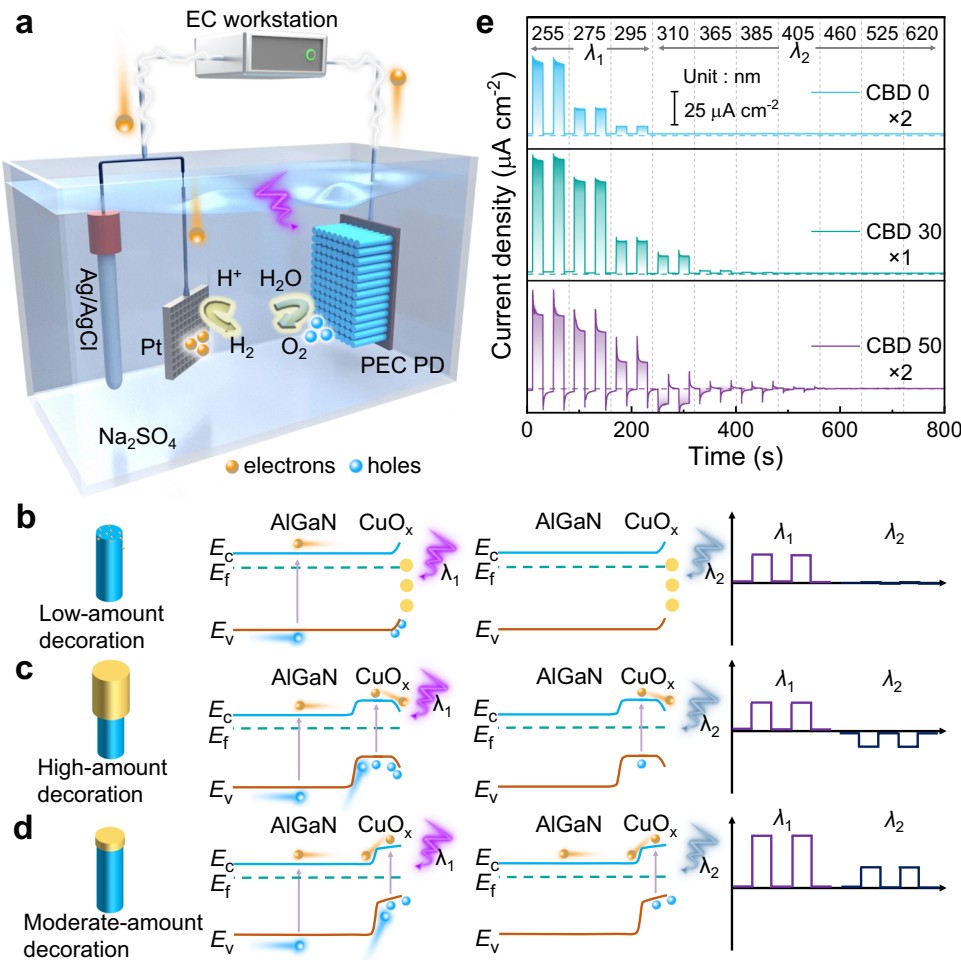

**Fig. 1 | Principles of AlGaN/CuO$_x$ nanowires for light-detection electrochemical cell. a** Construction and working principles of n-AlGaN nanowires-based light-detection electrochemical cell. **b**–**d** Wavelength-dependent operation model of the cell with low-level, high-level and moderate-level amount of decorated CuO$_x$, respectively. The energy band diagrams, CuO$_x$ decorated AlGaN nanowires and photocurrent signals are schematically shown. **e** Spectral response of AlGaN/CuO$_x$-nanowire-based PEC PDs at three different deposition times.

depositing $CuO_x$ composites, composed of CuO and $Cu_2O$ nanomixture. Regarding to conduction type of $CuO_x$, the AlGaN/$CuO_x$ heterostructure, indeed, could blur the direct evidence of the surface band bending. In order to get rid of the interfering from the heterojunction, the specific photoelectrode, with depositing $CuO_x$ layer on FTO substrate, was prepared for $I$–$t$ test to observe the polarity of the photocurrent generated by the bare $CuO_x$. When the bare p-type semiconductor contacts the electrolyte, the energy band at the semiconductor/electrolyte interface bends downward (Supplementary Fig. 2b), resulting in a negative photocurrent under UV light illumination. As shown in Supplementary Fig. 2c, the photocurrent with the negative polarity means the $CuO_x$ is the p-type semiconductor. According to our best knowledge, the operating mechanisms on the manipulation of carrier dynamics for the n-type III nitrides decorated by the $CuO_x$ could be analyzed through two types of role definition of the $CuO_x$ in the PEC process: as a surface co-catalyst to improve surface chemical dynamics[2] or as the p-type light-absorbing layer to construct the pn junction for inner structure optimazation[3,4].

When the n-AlGaN nanowires are decorated by a small amount of nano-structural $CuO_x$, the deposited oxides play the role of co-catalyst to promote the oxygen evolution reaction (OER). Therefore, the resulting positive-polarity photocurrent under deep UV light with $\lambda_1$ (defined as $E(\lambda_1) > E_g(AlGaN)$, where $E(\lambda_1)$ is the photon energy of $\lambda_1$ wavelength light and $E_g(AlGaN)$ is the bandgap of AlGaN) are improved, comparing to the bare AlGaN nanowires with a lower positive photocurrent, which is schematically shown in Fig. 1b. While under longer wavelength $\lambda_2$ light irradiation ($E_g(CuO_x) < E(\lambda_2) < E_g(AlGaN)$), where $\lambda_2$ is the wavelength of lower-energy light and $E_g(CuO_x)$ is the bandgap energy of $CuO_x$ composites), due to the excessively low amount of loaded $CuO_x$, there is no obvious photocurrent detected by electrochemical working station (Fig. 1b).

When the AlGaN is covered with a sufficient amount of $CuO_x$ nanocrystals (Fig. 1c), a compact thin capping layer was formed on the top of the nanowires. As an intrinsic p-type semiconductor for oxides of copper, the constructed pn junction on the basis of $CuO_x$/AlGaN contact should not be ignored in this case. When this p-$CuO_x$/n-AlGaN junction contacts the electrolytes, it is typically assumed that the energy band at $CuO_x$/electrolytes interface is commonly deemed to bend downwards by default, achieving electrochemical equilibrium at the p-type semiconductor/electrolyte interface. In this case, the direction of the built-in electric fields at the interfaces of $CuO_x$/AlGaN and $CuO_x$/electrolytes are opposite and, thus, there is competition on carrier transport at the two interfaces (Fig. 1c). When p-$CuO_x$/n-AlGaN nanowires are irradiated by high-energy $\lambda_1$ light, exhibiting the positive photocurrent for p-$CuO_x$/n-AlGaN nanowires. While irradiated by lower-energy $\lambda_2$ light, a negative photocurrent signal is generated under $\lambda_2$ light illumination. The detailed analysis on the transport of photogenerated carriers can be referred to in Supplementary Note 1.

In fact, besides of the described cases in Fig. 1b, c, there is another energy band structure for p-$CuO_x$/n-AlGaN nanowires in electrolyte, which is rarely mentioned before. Since the significant difference of work function between the $CuO_x$ and heavily n-doped AlGaN, the cladding layer of moderate amount of p-type $CuO_x$ could be completely depleted by n-type AlGaN. More than that, to further achieve the electrochemical equilibrium, the energy band would bend upward at the $CuO_x$/electrolyte interface to prevent the the electrons constantly transferring from the semiconductor to the electrolyte, which is shown in Fig. 1d. Although the $CuO_x$ is a p-type semiconductor, the downwards band bending is absent at semiconductor/electrolyte interface, which is replaced by abnormal upwards bending. From the viewpoint of the energy band characteristics, the upwards band bending of the reach-through $CuO_x$ layer perfectly matches with the band bending of p-$CuO_x$/n-AlGaN junction space charge region, significantly accelerating carrier transport and efficiently facilitating charge separation in the entire $CuO_x$ segment. Therefore, a large

photocurrent with positive polarity is detected by the illumination of $\lambda_1$ light. Similarly, when the p-$CuO_x$/n-AlGaN junction is exposed under $\lambda_2$ light, a small magitude of positive photocurrent generates in the circuit loop for this case, which is opposite to the direction of photocurrent for $\lambda_1$ light in case of Fig. 1b. In the other word, detecting the polarity switching of photocurrent between high- and lower- energy light is an effective method to confirm the truth of direction change of band bending at p-$CuO_x$/electrolyte interface, as the amount of deposited $CuO_x$ increases on the AlGaN nanowire. It is worth noting that the photoelectrode with the configuration of $CuO_x$/AlGaN/Si multi-heterojunction was designed for optimizing band alignment and carrier transfer. Generally speaking, as the results shown by the FDTD simulation (see below), most of UV light could be absorbed by the AlGaN/$CuO_x$-nanowire, which determines that a great many of the carriers are generated within the nanowires. Thus, for the upper $CuO_x$/AlGaN interface, it was assigned to photogenerated carrier regulation. For the lower AlGaN/Si interface, the heavily n-type doped Si substrates were adopted for plasma-enhanced molecular beam epitaxy (MBE) growth to achieve good electroconductivity between the n-AlGaN and Si. As shown in Supplementary Fig. 2c, due to the characteristics of the n-AlGaN/n-Si band alignment, either the electrons or holes separated from $CuO_x$/AlGaN interface can easily migrate across the AlGaN/Si interface and into Si substrate. It means that the AlGaN/Si interface plays the role as the linker for electric conduction.

In this work, the PEC light-detection devices based on the pristine AlGaN nanowires and the modified AlGaN/$CuO_x$ samples by chemical bath deposition (CBD) method for 10, 20, 30, 40 and 50 min are denoted as CBD-0, CBD-10, CBD-20, CBD-30, CBD-40 and CBD-50, respectively. Among these, the thickness of $CuO_x$ on AlGaN for CBD-10, CBD-30 and CBD-50 are ~3, ~20 and ~50 nm, respectively, which is supported by transmission electron microscopy (TEM) images in Supplementary Fig. 3. From this, the simulated energy band of AlGaN/$CuO_x$ heterojunction were obtained by the Silvaco TCAD (see "Methods") and are shown in Supplementary Fig. 4. It is not difficult to see that the tilt energy band of $CuO_x$ suggests the cladding oxides layer are totally depleted by the n-AlGaN nanowire for CBD-10 and CBD-30. Nevertheless, when the thick of $CuO_x$ layer increase to 50 nm, the energy band of $CuO_x$ become flattened at the far end of the junction and the the downward band bending could be established at the interface of un-depleted $CuO_x$/electrolyte. More importantly, the photocurrent on the spectral response of CBD-0, CBD-30 and CBD-50 in Fig. 1e shows the polarity-switching occurrence, which implies that the direction changes of band bending at p-$CuO_x$/electrolyte interface, from upwards for CBD-30 to downwards for CBD-50 as the increase of the $CuO_x$ thickness.

## Structural characterization of $CuO_x$/AlGaN nanowires

Si-doped n-type AlGaN nanowires were grown vertically on n-type silicon substrates by high-frequency MBE (see "Methods"). In this work, in order to enhance the photoresponse of the AlGaN nanowire-based PEC devices, surface decoration of $CuO_x$ nanostructures was involved on n-type AlGaN nanowires by CBD[20,21]. To elucidate the chemical properties and electronic states of the decorated AlGaN NWs, X-ray photoelectron spectroscopy (XPS) measurements were performed. In Fig. 2a, the XPS spectra of the CBD-modified AlGaN NWs shows typical peaks of Cu$2p_{3/2}$ and Cu$2p_{1/2}$, representing features of $Cu_2O$ and CuO, respectively[22,23]. Two binding energy peaks in Fig. 2a at 933.2 and 953.1 eV are associated with $Cu_2O$, while the other two smaller binding energy peaks at 934.6 and 954.5 eV are attributed to CuO. Additionally, the satellite peaks of Cu$2p_{3/2}$ and Cu$2p_{1/2}$ also confirm the existence of CuO on AlGaN. Since the peak shows proportionality to the number of copper atoms in their corresponding oxidation states, by calculating the ratio of peak areas, it is evident that the combined atomic ratio of $Cu^{2+}$ to $Cu^+$ are nearly one to one (1:1). On the one hand, the peak at 529.5 eV corresponds to the metal-oxygen

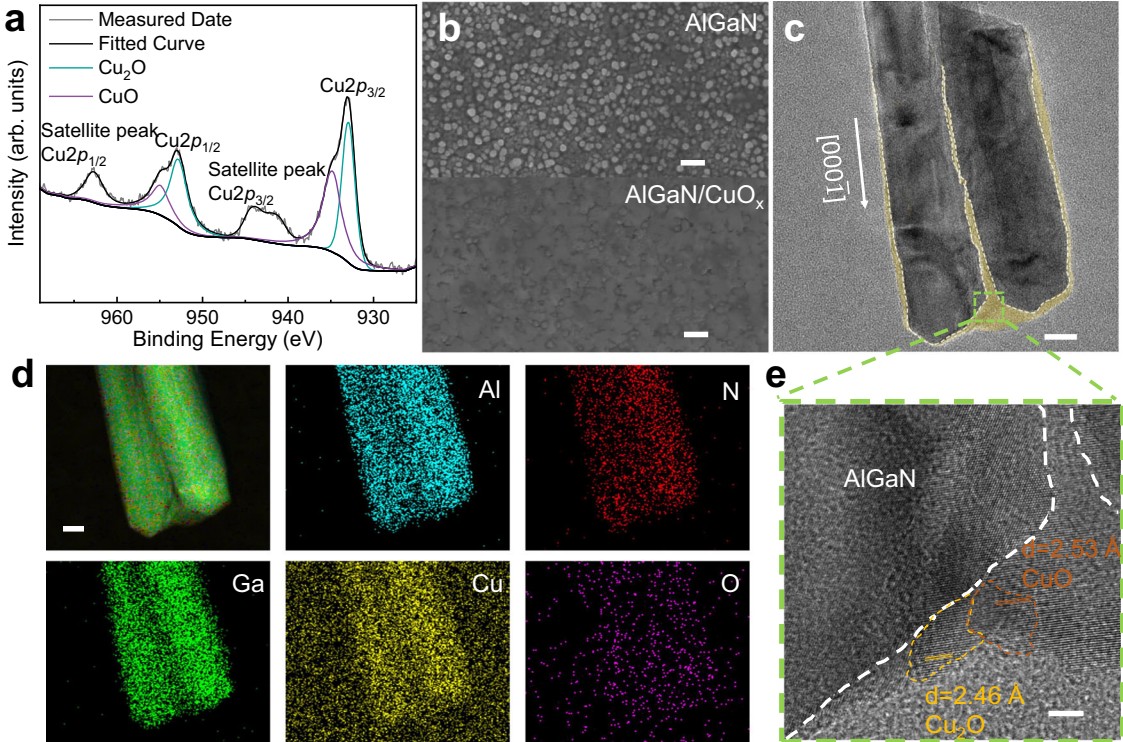

**Fig. 2 | Microstructure characterization of the AlGaN/CuO$_x$-based photoelectrode. a** The XPS peaks of Cu2$p$. **b** Top-view SEM images of the bare AlGaN (top; scale bar, 200 nm) and AlGaN/CuO$_x$ NWs (bottom; scale bar, 200 nm). **c–e** HRTEM image (**c**, scale bar, 20 nm) and elemental mapping (**d**, scale bar, 20 nm) of AlGaN/CuO$_x$ NWs (**e**, scale bar, 2 nm).

bonding; on the other hand, the peak locating at 531.6 eV could be attributed to surface hydroxide in the XPS spectrum of O1$s$ (see Supplementary Fig. 5)[22]. All of these XPS results demonstrate that the CuO$_x$ are well-structured on AlGaN nanowires by CBD method.

As shown in Fig. 2b and Supplementary Fig. 6, scanning electron microscopy (SEM) was conducted to characterize the changes on the morphology of AlGaN NWs before and after CBD deposition. In the top half of Fig. 2b, the morphology of the pristine AlGaN specimen shows that the as-grown AlGaN nanowires by MBE are vertically arranged on the substrate and have a uniform morphology in size. After CBD modification, the AlGaN nanowires are coated by a copper-oxide thin layer and the CuO$_x$ layer exhibiting the concavo-convex morphology, which is shown in the bottom half of Fig. 2b. Supplementary Fig. 6 provides the cross-sectional SEM images, where it is obvious that the CuO$_x$ layer resides atop the AlGaN NWs (~400 nm long). Furthermore, TEM image, shown in Fig. 2c, indicates that the more deposited CuO$_x$ mainly resides on the top of the nanowires and the side walls of the nanowires are covered with thin CuO$_x$ layer, forming a core-shell structure, which corresponds to the distribution properties of CBD deposition. In Fig. 2d, the delicate distribution of chemical composition is further characterized by scanning transmission electron microscopy (STEM) images and the corresponding energy spectrum element mapping. Figure 2e displays the high-resolution microstructure of CuO$_x$, in which the lattice fringes of Cu$_2$O and CuO are clearly visible. The stripes with a pitch of 2.46 Å correspond to the (111) lattice planes of cuprous oxide (JCPDS 005-0667), while the stripes with a pitch of 2.53 Å correspond to the (002) lattice plane of copper oxide (JCPDS 01-080-1268). In summary, these results confirm that CuO$_x$ are primarily loaded on the top as well as the side walls of AlGaN nanowires, forming AlGaN/CuO$_x$ heterostructures. This heterostructure enables efficient charge transfer at the interface and optimizes surface chemical reactivity, leading to the significant enhancement of photoresponse.

## Photoelectrochemical response enhanced by CuO$_x$ decoration

Surface decoration on n-type III-nitrides with metal oxides is an effective strategy to promote surface dynamics of target reaction and protect the III-nitrides from surface degradation[3,4]. Supplementary Fig. 7 depicts photoelectrochemical measurements of the original AlGaN nanowires and AlGaN/CuO$_x$ samples with different impregnation times under UV irradiation at a light intensity of 100 μW cm$^{-2}$. The photocurrent density results show that the current density first increased and then decreased with increasing CuO$_x$ modification time, showing that the 30-min CuO$_x$ deposition corresponds to the highest photoresponse. The photocurrent density of sample CBD-30 was recorded as 17.39 μA cm$^{-2}$ and is 6.1 times higher than that of the original sample (CBD-0). With increased deposition time, the amount of CuO$_x$ on the AlGaN nanowire surface increases, completing the construction of AlGaN/CuO$_x$ heterostructure. The photogenerated carrier separation is facilitated by the strong built-in electric field created by the reach-through band bending and the staggered-gap (type-II) band structure, as shown in Fig. 1d. However, when the CBD deposition time is prolonged to 40 min, the overloaded CuO$_x$ leads to downward band bending at CuO$_x$/electrolyte, causing severe carrier transport competition between the two interfaces and the decrease of current density (Fig. 1c).

Figure 3a shows the current–time ($I$–$t$) curves of PEC-PDs with sample AlGaN/CuO$_x$ nanowires (CBD-30) and the pristine AlGaN nanowires (CBD-0) at different light intensity illumination. It is not hard to identify that the AlGaN sample modified with CuO$_x$ nanostructures shows significantly improved PEC photoresponse compared with the bare AlGaN nanowires. Figure 3a, b shows that the photocurrent density ($I_p$) of the two samples are improved by the increasing light intensity, which implies the PDs can sensitively make response on the light power changing.

To more comprehensively evaluate the optical response of the two AlGaN-based PEC PDs, the calculated responsivity ($R$) and EQE are shown in Fig. 3b, c. In the first instance, the responsivity was adopted

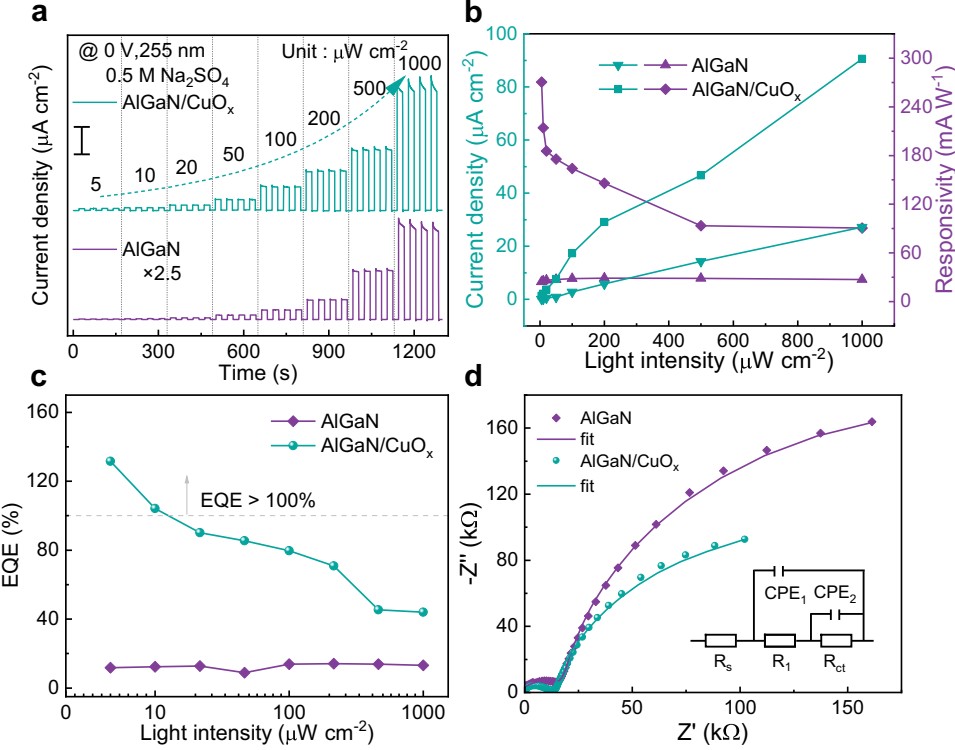

**Fig. 3 | Comparison of photoelectrochemical properties of AlGaN and AlGaN/CuOx PEC PDs under 255 nm illumination. a** Repeated on-off *I*–*t* characteristics of bare AlGaN and AlGaN/CuOx PEC PDs at different light power without bias. For making comparison more obvious, the *I*–*t* curve of the bare AlGaN PEC PD was enlarged by 2.5 times (scale bar, 20 μA cm⁻²). **b** Photoresponse characteristics (current density and responsivity) and **c** EQE of bare AlGaN and AlGaN/CuOx PEC PDs under 255 nm illumination at different light power. **d** EIS analysis of bare AlGaN and AlGaN/CuOx PEC PDs. The inset in (**d**) is the equivalent circuit model used to fit the Nyquist plots.

to characterize the sensitivity of the PEC PDs, which can be calculated through the equation:

$$R = \frac{I_p}{P} \qquad (1)$$

where $P$ is the incident light power intensity. The responsivity of the two PEC-PDs at 0 V bias and various light intensities are shown in Fig. 3b. The responsivity of the AlGaN/CuOx samples varied between 90.5 mA W⁻¹ and 270.6 mA W⁻¹, demonstrating excellent light sensitivity across a wide range of light intensity. In contrast, the maximum responsivity of the bare AlGaN PEC PD was only 29.04 mA W⁻¹ at 200 μW cm⁻². It is noted that, as shown in Fig. 3b, the responsivity of AlGaN/CuOx PD monotonously decrease with the light power intensity increase. The maximum responsivity of the AlGaN/CuOx PEC PD is obtained at the weakest light power intensity (5 μW cm⁻²), indicating that the AlGaN-CuOx co-catalyst has great potential for weak-light detection. Herein, it is important to introduce EQE to assess the photoelectric conversion efficiency:

$$EQE = \frac{hcI_p}{q\lambda P} \qquad (2)$$

where $c$ is the speed of light, $h$ is Planck's constant, $q$ is the electronic charge and $\lambda$ is the wavelength of incident light, respectively. As shown in Fig. 3c, the EQE of AlGaN/CuOx PD increases as the optical power intensity decreases, which is keeping with the results of responsivity. When the light intensity is 5 μW cm⁻², AlGaN/CuOx PD achieved an impressively high EQE of 131.5%, which was 11.1 times higher than the original PD (11.8%). Notably, due to the MEG effect triggered by the absorption of high-energy photons by CuOx, the EQE of AlGaN/CuOx

PD exceeds 100% without external bias. Within the range of light intensity from 5 to 1000 μW cm⁻², there is a significant change in the EQE of the AlGaN/CuOx PEC PD, dropping from 131.7% to 44%. In contrast, the EQE of the AlGaN sample shows no significant change and remains at a consistently low level. The main reason for this is that, compared with the bare AlGaN, the AlGaN/CuOx heterojunction sample, under the aid of MEG, generated many more photogenerated carriers at higher light power. These excessive carriers cannot be timely transferred away, thus lowering down the EQE at high light power. Besides of the responsivity and EQE, specific detectivity ($D^*$) is another crucial parameter for assessing the ability of PD to detect weak signals. When the noise is mainly dominated by the shot noise of the dark current ($I_{dark}$), the equation of the specific detectivity can be simplified as:

$$D^* = \frac{R\sqrt{S}}{\sqrt{2qI_{dark}}} \qquad (3)$$

Where $R$ is the responsivity of PD, $S$ is the area, and $q$ is the electron charge. The calculated $D^*$ values for different light intensities are shown in Supplementary Fig. 8. Impressively, the AlGaN and AlGaN/CuOx PEC PDs exhibit remarkable $D^*$ value of $5.56 \times 10^{11}$ and $6.17 \times 10^{12}$ Jones at 5 μW cm⁻², respectively, revealing the ability of AlGaN/CuOx PEC-type PD to detect weak light.

To evaluate the contribution of CuOx nanocomposites on PEC process, electrochemical impedance spectroscopy (EIS) measurement was employed under illumination of 255 nm light. EIS diagram could demonstrate the transport properties at the semiconductor/electrolyte interface. The diameter of EIS curve reveals the interfacial charge transfer resistance ($R_{ct}$), providing information about the charge transfer kinetics at the interface. The EIS plots of the bare AlGaN and

the CuO$_x$ decorated AlGaN samples are demonstrated in Fig. 3d. In order to quantitatively study the change of the $R_{ct}$, we employed an equivalent circuit that consists of a series resistance ($R_s$), a bulk resistance ($R_1$), a constant-phase element (CPE), and $R_{ct}$, which is shown in the inset of Fig. 3d. The value of $R_{ct}$, which decreased by more than an order of magnitude, dropped from $3.77 \times 10^6\ \Omega\,cm^2$ for AlGaN to $2.44 \times 10^5\ \Omega\,cm^2$ for AlGaN/CuO$_x$. The AlGaN/CuO$_x$ sample shows the smaller $R_{ct}$, which implies that the CuO$_x$ deposition can facilitate ultrafast charge transfer at the interface, benefiting the exciton extraction from CuO$_x$ layer.

## Photoresponse enhancement mechanisms

To elucidate the underlying mechanism on the enhanced PEC performance of AlGaN/CuO$_x$ nanowires assisted by the MEG effect, the behavior of charge generation and recombination were investigated using femtosecond transient absorption (TA) spectroscopy[24,25], in which low flux pulse excitation is designed to ensure that the number of excitons in the multiple photon excitation (MPE) state (i.e., absorbing multiple photons to create multiple excitons) can be neglected[15,26,27]. In order to prevent the Si substrate from interfering with absorption spectra, TA characterization were carried out on CuO$_x$ samples deposited on quartz substrates (see "Methods" and Supplementary Fig. 9). Supplementary Fig. 10 shows the time-resolved transient absorption spectra measured at a pump energy of 4.13 eV, a ground-state bleaching (GSB) peak appears near 520 nm under the excitation pulse, corresponding to the steady-state absorption peak of CuO$_x$, which is consistent with the result of UV-visible absorption spectrum (Supplementary Fig. 11a). In the meantime, the bandgap of CuO$_x$ composites is 1.9 eV, which is also confirmed by UV–visible absorption spectrum shown in Supplementary Fig. 11b. When a photon (with photon energy ≥3.8 eV) is absorbed by the CuO$_x$ nanostructure, a high-energy exciton is generated. In a very short time (~0.1 ps), the exciton decays via impact ionization, creating two electron-hole pairs (process I). The two hot electrons can then relax non-radiatively, e.g., via phonon emission (hot electron cooling), to form a ground-state biexciton (process II). Ultimately, the biexciton decays into a single exciton via Auger recombination[13] (process III). The entire evolution process of carrier dynamics is shown in Fig. 4a. The time-lapse spectral (Fig. 4b) shows strong peaks observed within the 1 ns, which gradually decrease over time, representing the thermal relaxation of multiple excitons. After 5 ns, the further decrease in peak value corresponds to the radiative recombination of the single exciton[28].

To determine the existence of MEG process and its quantum yield, TA spectroscopy probing the dynamic of ground-state bleaching is performed on spin-coated CuO$_x$ composite layer on quartz substrates. Since the TA instrumental response time (35 fs) is much shorter than the carrier lifetime (~1 ps), the influence of the instrument response is negligible, and it is feasible to the fit carrier dynamics curves using a biexponential function. Under small pump energy of $h\nu < 2E_g$ at low pump intensity, the GSB decay of CuO$_x$ only single exponential decay with a lifetime of >10 nm, indicating the neutral single-exciton recombination. The existence of MEG at higher photoexcitation energy (>$2E_g$) is evidenced by the emergence of fast decay with increased amplitudes. The lifetime of fast decay of 430 ps can be attributed to the biexciton Auger recombination. As shown in Fig. 4c, the quantum yield of CuO$_x$ at 4.13 eV ($2.12E_g$) high-energy photons is about 1.8, whereas it is only 0.98 at 3.4 eV ($1.74E_g$) low-energy photons, strongly evidencing the existence of over 100% quantum yield efficiency in CuO$_x$ nanostructure under high-energy ultraviolet light illumination. In the meanwhile, the efficiency of the MEG effect can be quantitatively characterized by carrier lifetime spectra, where the signal intensity is proportional to the number of excitons excited by the pump pulse in CuO$_x$[24,25,28]. In general, the calculated quantum efficiency is applied to quantitatively measure the MEG in a nano-semiconductor structure. As shown in Supplementary Fig. 12, the TA

signal rapidly decays from the initial peak to a steady value. The ratio of the peak signal strength $A$ to the platform signal strength $B$ is proportional to the average number of excitons generated in each stimulated nanosemiconductor. Therefore, at low flux incidence, the quantum efficiency of multiple exciton generation is expressed as[15,29]:

$$QE = \frac{A}{B} \times 100\% \qquad (3)$$

Where $A$ and $B$ are the absolute signal intensity of before and after Auger recombination. Here, as shown in Supplementary Fig. 13, it is worth noting that the multiphoton absorption events are excluded to ascertain the behavior of the photon absorption and further support the MEG effects.

Actually, besides the effective multiple exciton generation, efficient multiple exciton extraction and charge transfer promoted by the built-in electric field between the hetero-interface is crucial, which could effectively suppress the Auger recombination and take full advantage of the MEG effect to photoelectric gain in PEC process[17,19]. It is notable that the formed reach-through CuO$_x$ layer combined with the staggered-gap band structure of AlGaN/CuO$_x$ heterojunction, shown in Fig. 1d, is perfectly fit for extracting the multiplicative electrons from CuO$_x$ nanostructure into AlGaN. Once the multiple excitons were generated by high-energy photons in the CuO$_x$ layers, an appropriate method of extracting charges is very crucial for suppressing thermal relaxation, due to the short lifetime of the multiple excitons. The electric field in the reach-through CuO$_x$ layer can impel the charge separation effectively. The significance of reach-through CuO$_x$ layer is that photo-induced carriers can be impelled to drift across the whole depleted layer by the built-in electric fields between AlGaN and electrolyte, yielding much higher efficiency of charge extraction than diffusion movement. Since then, the following issue should be emphatically considered is that the characteristics of spatial distribution on optical field and the photogenerated multiple excitons. In Fig. 4d, when the pristine AlGaN NWs are irradiated by the light with longer wavelength (>300 nm), there are no obvious photoresponse to be detected. Only under the photoexcitation by the short-wavelength light (<300 nm), an observable photocurrent occurred, which is attributed to the intrinsic band-to-band carrier transition in AlGaN. By contrast, within the region of 255–365 nm, the more prominent photoresponse of AlGaN/CuO$_x$ PD could be attributed to the excited CuO$_x$ coating layer. The most fascinating enhancement of photoresponse for the AlGaN/CuO$_x$ PD appears under <300 nm UV light irradiation, which is ascribed to the participation of MEG. Here, it needs to be mentioned that the AlGaN/CuO$_x$ PD, there is no photocurrent can be observed under longer than 400 nm light irradiation, though the photon energy is exceeding the bandgap of CuO$_x$. The EQE of bare AlGaN and AlGaN/CuO$_x$ PEC PDs under different light wavelengths (255–620 nm) was also charted in Supplementary Fig. 14. It is clearly indicated that EQE of AlGaN/CuO$_x$ PD exceeds 100% irradiated by UV light below 300 nm. Specifically, the photon energy dependence of the distribution of electric field in the nanowires should be deeply considered, which totally determines the spatial distribution of photogenerated carriers. According to the results of the finite-difference time-domain (FDTD) simulation (the numerical model can be referred to Supplementary Fig. 15), it can be explicitly recognized that, under short-wavelength light illumination, the distribution center of the optical field as well as its induced photogenerated carriers locate at within CuO$_x$ layer as well as the vicinity of AlGaN/CuO$_x$ interface. However, as the light wavelength increases from 255 to 385 nm, the strongest absorption region shifts downwards, from AlGaN/CuO$_x$ interface into the bulk of AlGaN nanowires, as shown in Fig. 4e–j. In other words, in the short-wavelength light irradiation, the electron-hole pairs generated could be immediately swept into built-in electric field of the AlGaN/CuO$_x$ interface, expediting the carrier separation

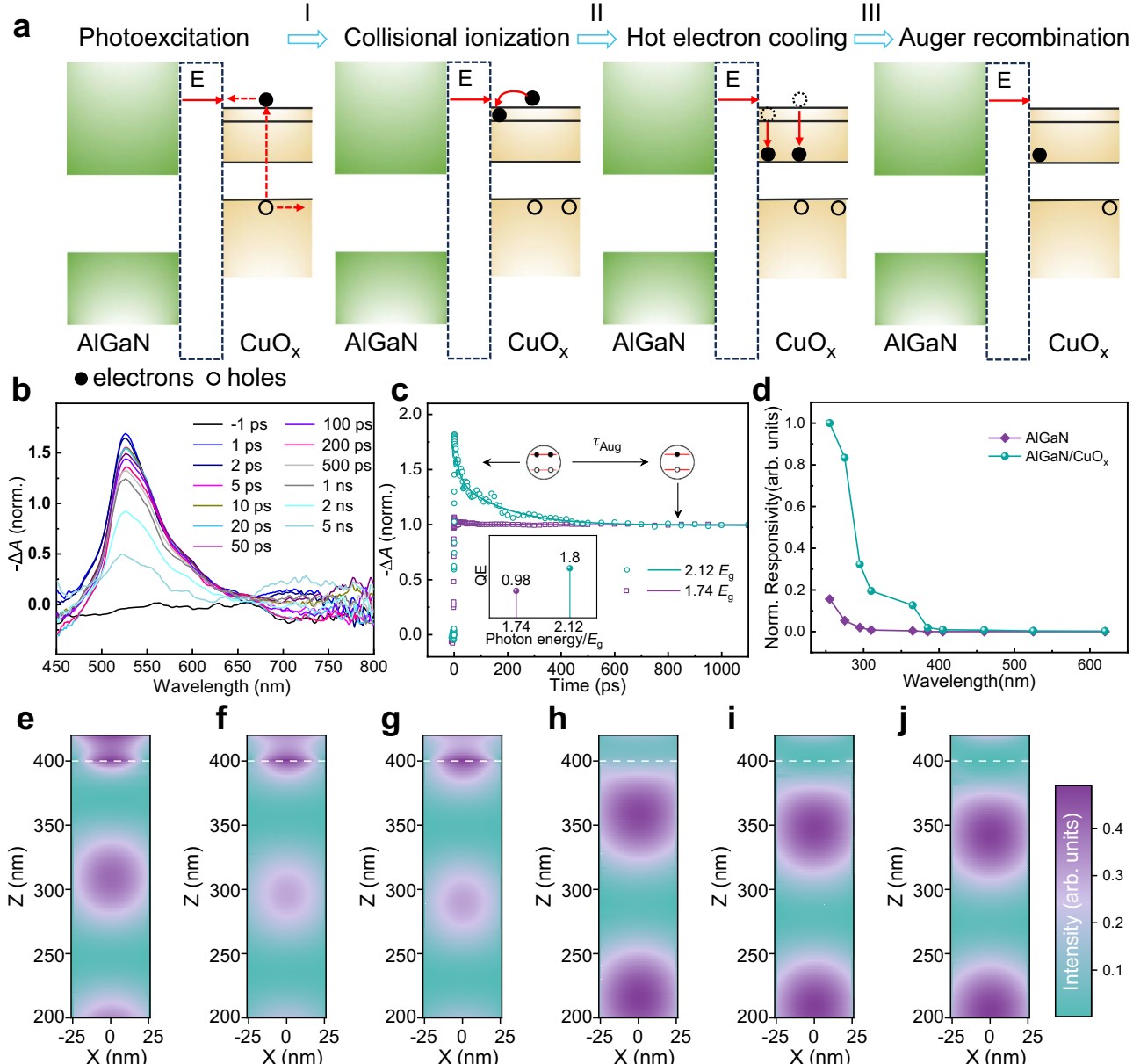

**Fig. 4 | The mechanism and characterization of the synergy mechanism of MEG and charge-transport kinetics. a** The process of MEG effect. **b** The TA spectra showing the first exciton bleach of CuO_x at different delay time. The pump is at 300 nm with a pulse fluence of 10 nJ cm⁻². The $A$ represents the absorption for TA measurement. **c** Bleaching dynamics with and without MEG in CuO_x ($E_g = 1.9$ eV). The lines are a two-exponential fit that indicates the presence of the short-lived bi-

excitonic component. Inset: MEG QE in CuO_x as a function of pump photon energy normalized by $E_g$. **d** Normalized responsivity as a function of wavelength for the bare AlGaN and AlGaN/CuO_x PEC PD at 0 V bias. Simulated electric-field intensity distribution of AlGaN/CuO_x NWs at 255 nm (**e**), 275 nm (**f**), 295 nm (**g**), 310 nm (**h**), 365 nm (**i**) and 385 nm (**j**) wavelength. The white dotted line represents the interface between AlGaN and CuO_x.

and then resulting in detectable photocurrent. On the contrary, under long-wavelength light illumination, the generated carriers occur in the bulk of the AlGaN nanowires, keeping away from the top of the nanowires and failing to make a contribution to the photocurrent. Moreover, the optical absorption efficiency of CuO_x at 255 nm is 2 times higher than at 400 nm through comparison. In General, under long-wavelength light, such fast relaxation process impedes the application of MEG theory to PEC photodetection for enhancing external quantum efficiency[28]. By manipulating the spatial distribution of photogenerated carrier and combining with the effects of the strong built-in electric field in AlGaN/CuO_x interface, the multiplied carriers in the CuO_x layer can be effectively separated and transferred without cooling, enabling an efficient MEG process[14,17,19,30] and achieving 131.5% external quantum efficiency.

## High-resolution deep-ultraviolet imaging

Because the PEC-based device involves both physical and chemical processes, it offers diverse methods for adjusting light-responsive behavior. Figure 5a–c demonstrates the sensitivity of the light response to the bias potential and electrolyte concentration. When the bias potential was adjusted between −0.2 and 0.8 V, the photoresponse is enhanced from 16.5 to 48.3 µA cm⁻², which is attributed to the acceleration on carrier separation and transport in the photoelectrode by bias potential[3,4]. In addition to the external bias, the photocurrent of AlGaN/CuO_x PD also depends on the electrolyte concentration. As the electrolyte concentration increases, the photocurrent density increases at first and then decreases. When the electrolyte concentration is 0.5 M, the $I_p$ reaches the maximum value. However, at higher ion concentrations than 0.5 M, redox at the

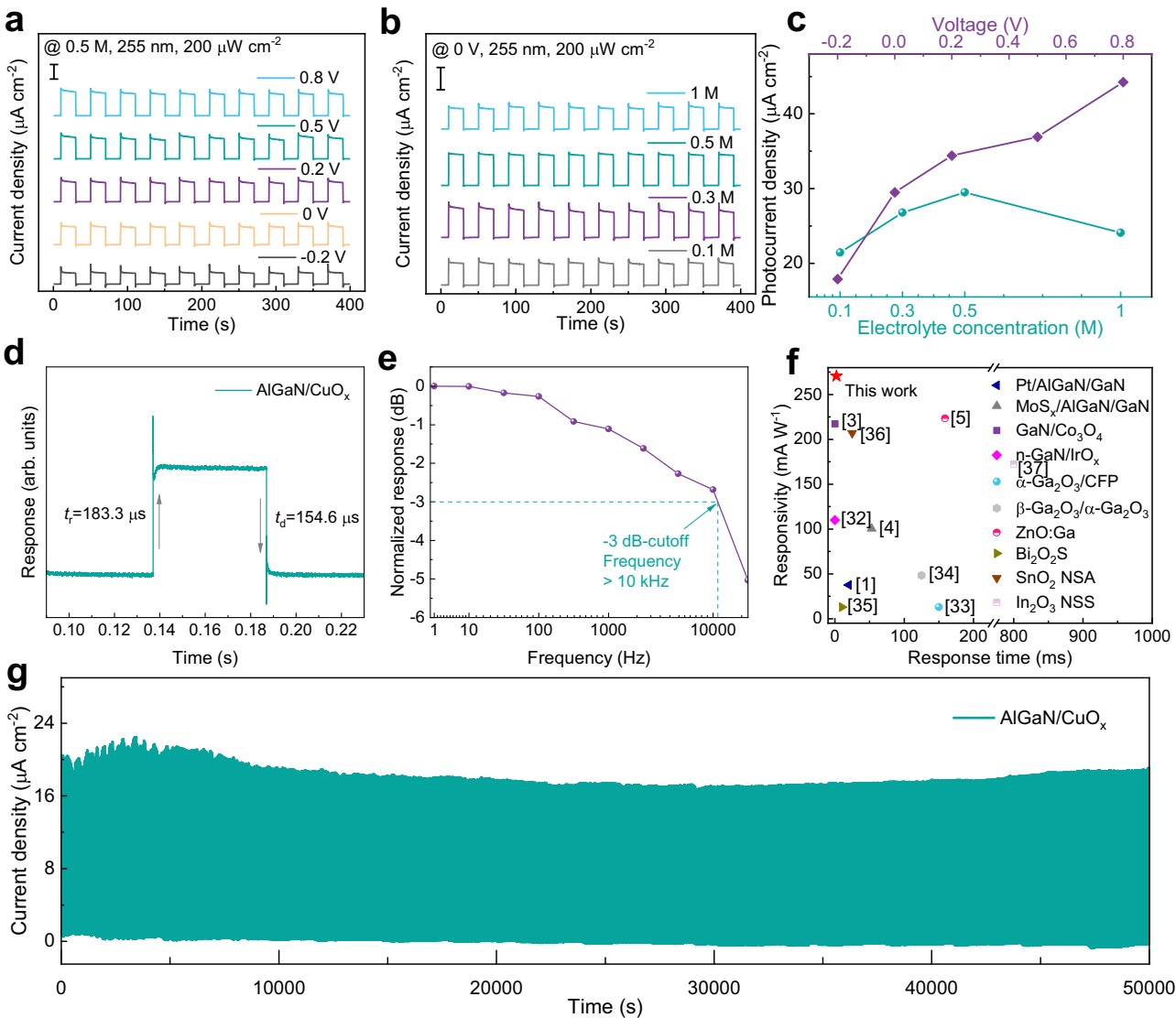

**Fig. 5 | High PEC properties and stability of AlGaN/CuO_x PEC PD. a–c** $I$–$t$ curve and photocurrent density of AlGaN/CuO_x PEC PD under different electrolyte concentration (0.1 M, 0.3 M, 0.5 M and 1 M; **a** scale bar, 20 µA cm$^{-2}$) and bias voltage (−0.2 V, 0 V, 0.2 V, 0.5 V and 0.8 V; **b** scale bar, 20 µA cm$^{-2}$). **d** The rise and decay time for AlGaN/CuO_x PEC PD. **e** The frequency response curve with −3 dB cut-off frequency. **f** Comparison of responsivity and response time between AlGaN and AlGaN/CuO_x PEC PD. **g** In all, 50,000-s stability test of AlGaN/CuO_x PEC PD.

interface is hindered, resulting in a decrease in photocurrent[4]. This phenomenon can be attributed to the adsorption of high-concentration $SO_4^{2-}$ ions onto the photoanode surface, which occupies active sites and impedes hole transfer to reactants. In addition, the localized accumulation of oxidation products ($O_2$ and $H^+$) induces proton overconcentration, together suppressing oxidation kinetics.

To display the light response speed of the PD, Fig. 5d (AlGaN/CuO_x) and Supplementary Fig. 16 (AlGaN) show the response time ($t_r$) and decay time ($t_d$) intervals of the photocurrent curve at a bias potential of 0 V under 255 nm illumination. The $t_r$ and $t_d$ are defined as intervals from 10 to 90% of the maximum photoelectric values and from 90% to 10%, respectively. The corresponding $t_r$ and $t_d$ of AlGaN/CuO_x are 183.3 and 154.6 µs, respectively, while the response speeds of the bare AlGaN nanowires were 360.8 and 241.3 µs, respectively. It is not difficult to see that the improved response speed of AlGaN/CuO_x PD is most likely due to the accelerated charge transfer process and redox rate at the interface. To thoroughly study the frequency characteristic of the AlGaN/CuO_x PEC PD in electrolyte, frequency-based tests were conducted, with presenting the time response of the PEC PD in the 1 Hz–20 kHz

frequency range in Supplementary Fig. 17. It can be observed that the output waveform remains stable as a square wave with clear rising/falling edges up to 1 kHz, yet when the frequency increases to 10 kHz, the output waveform transitions from a square wave to a triangular wave, probably because the PD reaches its response-speed limit and lacks sufficient time to stabilize at high frequencies. The frequency-response curve (Fig. 5e) was obtained by measuring the amplitudes of output signals at different frequencies, enabling to estimate that the −3 dB cut-off frequency of the AlGaN/CuO_x PEC PD in sodium sulfate solution is higher than 10 kHz. In our opinion, although achieving ultra-high-frequency response for PEC devices is still a challenge, a few promising strategies were proposed, such as interface engineering to eliminate recombination, nanostructure design to reduce the carrier diffusion path, circuit optimization for impedance matching. Figure 5f shows the comparison on the performance of the prepared AlGaN/CuO_x PD with the recently reported PEC photodetectors[1,3–5,31–36]. The AlGaN/CuO_x PD, in our work, shows more superior photoresponse performance than the other listed works in Fig. 5f, with the responsivity of 270.6 mA W$^{-1}$ and EQE of 131.5%.

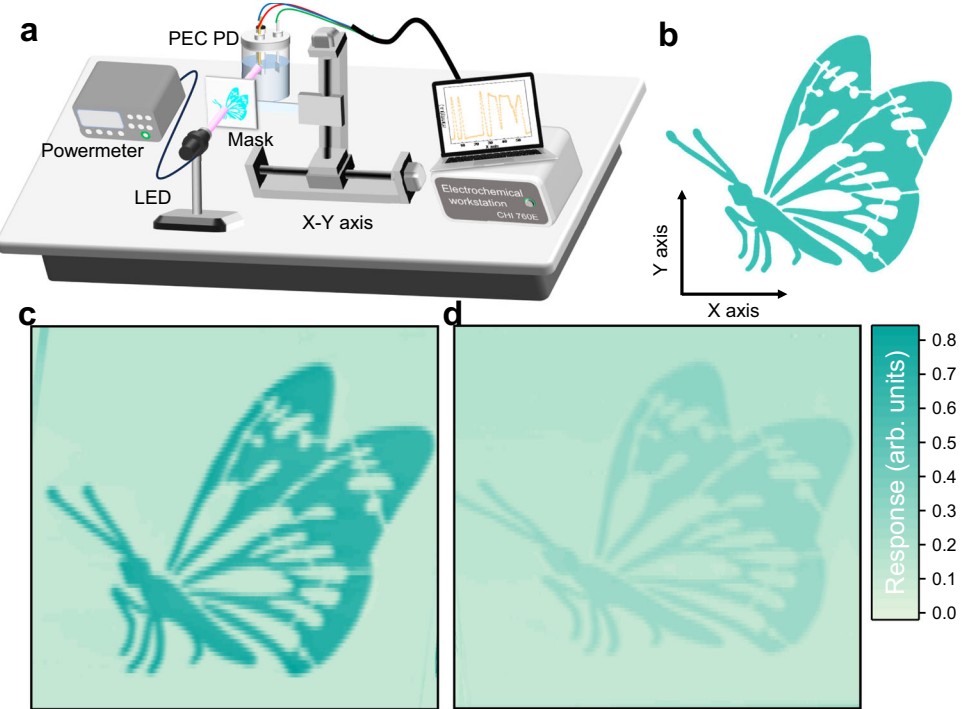

**Fig. 6 | Imaging with the AlGaN/CuO$_x$ PEC PD. a** Schematic illustration of set-up for imaging. **b** The pattern of the mask. **c**, **d** Imaging comparison of AlGaN/CuO$_x$ and bare AlGaN PEC PDs.

Generally speaking, the durability and long-term stability of multicycle photodetectors are the crucial characteristics for further application. Figure 5g shows the 50,000-s long-term on/off response of AlGaN/CuO$_x$ samples under illumination of 200 μW cm$^{-2}$ without bias. The photocurrent of AlGaN/CuO$_x$ samples only exhibits slight attenuation after 50,000 s of long-term operation, highlighting the remarkable stability of AlGaN/CuO$_x$ samples in multiple cycles. Actually, because of the surface passivation by deposited CuO$_x$, the AlGaN nanowires were substantially mitigated from the photo-induced corrosion, resulting in excellent durability and stability for AlGaN/CuO$_x$ PD. And more notably, the main reason for selecting Na$_2$SO$_4$ electrolyte is that it could create a relatively stable ionic environment, suppressing interfering with the photoelectrochemical (PEC) process by the side reactions. In addition, though the alkaline electrolyte could provide higher OH$^-$ ion concentration and might improve the photoresponse, the original Na$_2$SO$_4$ is not suitable to be replaced with the alkaline electrolyte, in consideration of the chemical stability of PEC PD. Alkaline electrolytes might corrode the surface of nanowires during PEC testing, thereby reducing the long-term stability of PEC PDs. As shown in Supplementary Fig. 18, the photoresponse performance drops by over 50% within the initial 12,000 s.

Finally, the excellent photon response of the AlGaN/CuO$_x$ PEC PD to specific wavelengths was further substantiated through monochromatic light imaging. Figure 6a illustrates the component schematic, where the light source is provided by a light-emitting diode, and a "butterfly" pattern mask (Fig. 6b) is positioned between the light source and PD. The mask, having a subtle pattern with a fine outline, is capable of scanning in both the X and Y axes via a displacement platform. As shown in Fig. 6c, owing to the superior responsivity of the AlGaN/CuO$_x$ PEC PD at 255 nm compared to the AlGaN PEC PD, a distinct "butterfly" pattern is discernible with the AlGaN/CuO$_x$ PEC PD at a wavelength of 255 nm. More details about the imaging tests can be found in Supplementary Fig. 19. In contrast, a significantly more blurred pattern is observed with the original AlGaN PEC PD (Fig. 6d). As previously mentioned, the remarkable contrast in image clarity between the two devices further corroborates the ultra-high

responsivity of the AlGaN/CuO$_x$ PEC PD under weak UV illumination. The high-responsivity PDs facilitate high-resolution image acquisition in dynamic environments.

Because of the high photoresponse performance, it is feasible to discuss the proposed PEC PD for further application and even for commercialization in the future. Totally, there are still several issues to be solved, such as device packing and system integration. At present, from the perspective of the authors, two technical issues need to be addressed. Because of no external supply, the photoresponse of the self-powered PEC PD should be improved further. Certainly, the MEG involved PEC PD in this paper provides one effective solution. For another thing, the liquid electrolyte as well as the bulky vessel impedes the commercial progress. In fact, the traditional water and electrolytic vessel for the PEC PD could be replaced by the solid electrolyte-based capsule, which significantly reduces the volume of PEC PD, less than several hundredths of the original PEC cells. It greatly improves the practicality in the field of portable and wearable electronics[37,38].

## Discussion

In summary, up to now, the quantum efficiency of MEG is still inefficient in optoelectronics because the incident light energy must be much greater than the $2E_g$ of the excited nano-semiconductors to generate sufficient excess carriers. Recently, optimizing energy band structure and enhancing the internal electric field intensity has been proved to be a promising strategy to extract excitons generated by the MEG effect and improve the photoelectric conversion efficiency[17,19].

We report a PEC device with high responsivity and external quantum efficiency of over 100% after appropriate CuO$_x$ composite modification, providing the reach-through band bending on the AlGaN nanowire surface by CBD method. By manipulating spatial distribution of optical fields, the MEG occurs within the whole depleted CuO$_x$ layer with tilt energy band, the generated multiple excitons can be instantaneously separated by the built-in electric field against the exciton Auger recombination. The separated electrons migrate to the interface of AlGaN/CuO$_x$ and are swept into the AlGaN by the strong built-in electric field of the type-II heterojunction; the holes, at the same time,

transfer to the $CuO_x$/electrolyte interface to participate in the OER at surface. From this, comparing to the bare AlGaN PEC PD, the EQE of the device based on AlGaN/$CuO_x$ heterojunction significantly increased from 11.8 to 131.5%, with an 11.1-fold increase in responsivity of 270.6 mA W$^{-1}$ and a fast response speed under 255 nm light illumination. Notably, both the photoresponsivity and quantum efficiency achieve high-level results among self-powered PEC PD reported to date. This work proves the necessity of precise energy band designing for realizing the carrier multiplication from MEG, opening up the insight into constructing self-powered PEC-based optoelectronics in the future.

## Methods

### Growth of nanowires and $CuO_x$ nanocomposite
AlGaN nanowires were grown on Si substrates using plasma-assisted MBE under nitrogen-rich conditions. To remove organic contaminants and surface oxides from the silicon substrate, it is thoroughly cleaned sequentially with methanol, acetone, and hydrofluoric acid. After cleaning, the substrate was installed in the MBE chamber. Before the growth process, the Si substrate was first annealed at 860 °C for 20 min to remove the remaining oxides and restore the growth front. The flows of Al, Ga, Si, and Mg are controlled by a melt pool, with nitrogen ions provided by a radio frequency plasma source. The AlGaN nanowires were grown at 780 °C for 0.5 h.

Subsequently, $CuO_x$ was grown using chemical bath deposition. 0.5 M $CuSO_4$ was dissolved in 10 mL of deionized water to form a solution. Then, 40 mL of 0.5 M $Na_2SO_3$ aqueous solution was slowly added to create mixed solution I, containing $SO_4^{2-}$ and $Cu^{2+}$ ions. The prepared AlGaN samples were immersed in solution I and stirred for 15 min. Then, 0.25 M NaOH aqueous solution was added to the mixed solution at 70 °C for 30 min. The sample was then removed, washed three times with deionized water, and dried at 70 °C in a vacuum oven. Finally, AlGaN samples with $CuO_x$ nanocomposite structures were obtained. Similarly, immerse the cleaned quartz substrate into Solution I. Under the same conditions, a sample of the $CuO_x$ nanocomposite structure supported on the quartz substrate will be obtained.

### Characterization of nano-structure
The morphology and microstructure of AlGaN and AlGaN/$CuO_x$ nanowires were characterized using a field emission scanning electron microscope (FESEM, Hitachi Regulus 8220) and a transmission electron microscope (TEM, Talos F200s). The chemical states of the elements in the samples were evaluated using an X-ray photoelectron spectrometer (XPS, Thermo Scientific, Escalab 250xi) with a monochromatic Al kα source (15 kV, 150 W). The position of the binding energy peak was calibrated by the C1s peak (284.6 eV). The carrier dynamics curves of $CuO_x$ nanostructures with an excitation wavelength of 300 nm were obtained using CEL-TAS3000 femtosecond transient absorption spectroscopy. Transient absorption spectroscopy was carried out by using an optical instrument combined a frequency-doubled mode-locked Ti/sapphire femtosecond laser (coherent) and an optical parametric amplifier system. The amplified Ti/sapphire femtosecond laser generates seed pulses with a 35 fs pulse width and a repetition rate of 1 kHz. The seed pulses are divided into two distinct beams. The strong beam is sent to the OPA system and provided the 300 nm pump laser pulse, and the other one is focused onto a nonlinear crystal to generate a white light continuum, providing broadband of 400–800 nm UV–vis probe light. The excitation beam has a low energy of 10 nJ cm$^{-2}$ per pulse to avoid the exciton–exciton and exciton–charge annihilation effects.

### Fabrication and measurements of photoelectrodes
AlGaN and AlGaN/$CuO_x$ samples were prepared as photoelectrodes for PEC testing. The natural oxides on the back of the silicon are first removed and then the substrate is coated with a Ga-In alloy to ensure ohmic contact. The sample is then attached to a copper strip using conductive silver slurry. Then, in addition to the nanowires required for light detection, the photoelectrodes are coated with insulating epoxy resin. Finally, the photoelectrodes were dried for 24 h before measurements.

To record the photoresponse properties of the photoelectrode, a three-electrode system was used, consisting of the photoelectrode (working electrode), Pt (counter electrode), and Ag/AgCl (reference electrode). Four different concentrations of $Na_2SO_4$ (0.1 M, 0.3 M, 0.5 M and 1 M) were used as the electrolyte in a quartz cell with high UV transmittance. Apart from taking into account the convenience and biosecurity of experimental operation, the long lifespan of LED light source is conducive to long-term stability characterization. The photoelectrodes were exposed to 255 nm LED light with varying light power intensities, and current–time, and electrochemical impedance spectroscopy (EIS) were analyzed on an electrochemical workstation (CHI 760E, Shanghai Chenhua). The incident light was provided by an array of light-emitting diodes with wavelengths of 255–620 nm. The intensity of the light involved in the experiment were calibrated in real time by an UV radiometer (LS-125) and a solar power meter (LH-122). During the light calibration, a real-time and mature protocol was implemented.

### Finite-difference time-domain simulations
FDTD calculations with commercial software (FDTD solution, Lumerical Inc., Vancouver, Canada) were used to simulate the optical properties of AlGaN/$CuO_x$ nanostructures. To ensure the accuracy of electric field calculations, the sizes of the AlGaN and $CuO_x$ in the simulation models are based on the SEM and TEM results. To simplify the simulations, the shape of AlGaN and $CuO_x$ is set to be a rectangle with 50 × 400 nm and 20 × 50 nm, and the substrate is set as Si. The light source is a plane wave. The refractive index of the AlGaN and $CuO_x$ nanostructure was taken from the reported values[39,40].

### Simulation of semiconductor energy band
Silvaco TCAD, based on the ATLAS device simulator, was adopted to analyze the energy band structure of AlGaN/$CuO_x$ heterojunction NWs. Poisson's equation and the continuity equation were used in the numerical procedures. For the III-nitride semiconductors, the spontaneous and piezoelectric polarization effects were both taken into account for the polarization physical model.

## Data availability
The data that support the findings of this study are available in the article, supplementary information file, source data file or from the corresponding authors upon request. The refractive indices of the AlGaN and $CuO_x$ nanostructure used for simulations were taken from the reported values[39,40] and are included in the source data file. Source data are provided with this paper.

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

## Acknowledgements

This work was funded by the National Key Research and Development Program of China (2022YFB3605400); National Natural Science Foundation of China (62374094, 62104110, and 62174016); Project funded by China Postdoctoral Science Foundation (2023T160332); Training Plan for Principals of Key Projects in Nanjing University of Posts and Telecommunications (NY224084); Basic Research Pilot. Project of Suzhou (SJC2022004).

## Author contributions

J.X., J.W., R.Z., G.Y. and D.C. developed the idea and designed the experiments. P.S., X.W., G.X., X.T., T.P., B.L. and K.W. performed the MBE growth and materials characterization. X.W., G.X., X.T., T.P., Z.C., Q.C. and Z.H. measured the photodetection experiments and analyzed the data. J.X., T.Z. and X.W. co-wrote and revised the paper. All the authors discussed the results and commented on the manuscript.

## Competing interests

The authors declare no competing interests.

## Additional information

**Supplementary information** The online version contains
supplementary material available at

Pengfei Shao, Guofeng Yang, Dunjun Chen or Jin Wang.

**Peer review information** *Nature Communications* thanks the anon-
ymous reviewer(s) for their contribution to the peer review of this work. A
peer review file is available.

