## [Transparent Peer Review file · Nature Communications]

Multiple exciton generation boosting over 100% quantum efficiency photoelectrochemical photodetection

Corresponding Author: Professor Junjun Xue

Version 0:

Reviewer comments:

Reviewer #1

(Remarks to the Author)

The manuscript presents an interesting study on the MEG-assisted photoelectrochemical photoresponse that leads to carrier multiplication and quantum efficiency enhancement. This manuscript provides an approach to manipulating surface energy bands to extract carriers from the site of exciton generation. There are a few minor issues that need to be addressed to improve the manuscript further. Below, I outline my specific comments and suggestions.

Comments:

1. In performance evaluation, light intensity calibration is important. The authors need to provide details on how the light intensity was calibrated.
2. The title refers to "photocatalytic photodetection," while the text uses "photoelectrochemical." Which term is more appropriate?
3. In Figures 3b and 3c, why does the EQE of the AlGaIn/CuOx sample decrease with increasing light intensity? At 1000 $\mu\text{W}/\text{cm}^2$, the EQE is only 40%, much smaller than at 5 $\mu\text{W}/\text{cm}^2$, while the EQE of the AlGaIn sample does not change significantly. Also, would the EQE be even larger at lower light intensities?
4. The authors only calculated the EQE at 255 nm. They should also calculate EQE at other wavelengths to further support the MEG effect.
5. In Figure 5b, the change in Na₂SO₄ concentration of the electrolyte affects performance. As the authors mention, higher concentrations improve the conductivity of the electrolyte. However, why does a higher Na₂SO₄ concentration lead to an excess of proton concentration? Why does a higher Na₂SO₄ concentration inhibit the oxidation half-reaction at the photoanode/electrolyte interface? The authors should provide further explanation.
6. Regarding the oxidation half-reaction: $\text{OH}^- + \text{h}^+ \rightarrow \text{OH}^*$, in the Na₂SO₄ solution used in the manuscript, which is neutral, there are not large quantities of OH^- . The authors should clarify this point.
7. The vertical axis in Figure 4b should be corrected.
8. The authors should further provide SEM cross-sectional images of the nanowires before and after loading.
9. The authors should provide the photocurrent response curve of Figure 6c.
10. As said by authors, the "reach-through" layer is important for extracting excitons. Is the electric field in the completely depleted CuOx layer crucial? Please make comments on this.
11. To determine the existence of the MEG process and its quantum yield, TA spectroscopy probing the dynamic of ground-state bleaching is performed on a spin-coated CuOx composite layer on quartz substrates. Here, why did the authors use different samples for testing? Whether the growth condition are consistent with the above article, whether the sample has been confirmed by other characterization methods.
12. The author calculates that the external quantum efficiency of AlGaIn/CuOx PEC PD is 131.5% , and the quantum yield of CuOx is about 1.8 at 4.13 eV high-energy photons, both the concepts of QE and EQE are mentioned. Should emphasize the connection between the two and the numerical differences? What is the relationship between them?

Reviewer #2

(Remarks to the Author)

This manuscript reports an AlGaIn/CuOx self-powered photodetector with a record-high responsivity of 270.6 mA W⁻¹ and an external quantum efficiency of 131.5% due to multiple exciton generation. While the experimental methodology is technically

rigorous, several critical scientific problems remain unclear, including the fundamental working mechanisms of the PEC photodetector and the photophysical process underlying the tunable photoresponse. Furthermore, this PEC photodetector demonstrates a relatively slower response and weaker stability compared to previously reported self-powered 2D PEC detectors. Additionally, there are significant issues with English grammar and formatting in the manuscript. Therefore, I do not believe the manuscript is suitable for publication in Nature Communications at the present stage. However, I would encourage you to resubmit a new manuscript after extensive revision.

1. The experimental evidence of p-type CuOx and n-type AlGaIn semiconductors lacks, which are key results to realize a negative and positive photocurrent signal and demonstrate the interfacial carrier transportation process.
2. AlGaIn/CuOx-nanowire is grown on n-Si (111) substrate, forming Si/AlGaIn/CuOx heterostructure. Certainly, the multi-interfacial electron transfer process also plays significant role in improving photodetection performance. However, the influence of the substrate and interface carrier transfer on this photodetector has yet to be explored and studied.
3. The author pointed out that the working principle of the PEC photodetector is driven by an oxidation reaction ($\text{OH}^- + \text{h}^+ \rightarrow \text{OH}^*$) and a reduction reaction ($\text{OH}^* + \text{e}^- \rightarrow \text{OH}^-$). However, it is unclear how the author confirmed this specific chemical reaction. Previous reports have indicated that the water redox reaction, which generates H₂ and O₂, also contributes to photocurrent generation.
4. Since OH⁻ can consume the photogenerated holes, why did not select alkaline electrolyte in the PEC measurement to further improve photoresponse?
5. Why the light intensity-photocurrent is nonlinear trend in Figure 3b? However, this nonlinear response is disadvantage for practical application of photodetector. Please comment on whether such kind of photodetector could become commercialized?
6. Why does the current in Figure 5g suddenly surge during the moment of illumination, while that of Figure 4(a) and 5(a) does not?
7. In Fig 3d, the Nyquist plot should be fitted with equivalent circuit model to better understand the carrier dynamics.
8. The 3000-second photocurrent test appears to inadequately reflect the stability performance of the PEC photodetector. Therefore, it is recommended that the photocurrent measurement time should exceed 30000 seconds or longer.
9. The specific detectivity and -3 dB cutoff frequency are also another important parameter for evaluating the device performance and should be calculated.
10. The response time of AlGaIn/CuOx PEC photodetector is in millisecond scale, which is far slower than previous reported microsecond-level photodetector. The measured result may be incorrect and influenced by machine response time of electrochemical workstation. It is suggested that the oscilloscope is used to collect the signal in the frequency range of 1–100 kHz.
11. In the FDTD simulation, AlGaIn and CuO are modeled with rectangular structures, whereas their actual morphology is that of core-shell nanowires. Might this simplification impact the accuracy of the simulation results?
12. It is recommended that the authors add the application potential of the research (e.g., UV imaging) at the end of the abstract to highlight the practical significance.
13. The author employed transient absorption technology in the manuscript to measure the carrier dynamics at corresponding locations in the CuOx and fitted the data using a biexponential function. In practice, when fitting carrier dynamics curves, the influence of instrumental response on the experiment generally needs to be considered.
14. Could you elaborate on the reasoning behind choosing LED lamps over lasers, which possess superior monochrome and directional properties, for use in the optical imaging experimental device? Furthermore, it is necessary to provide details regarding the processing method for optical imaging data, including the software and algorithms used, as well as the specifications for the size of the hollow part of the mask, the size of the laser spot, and the scanning speed within the optical imaging system.
15. The article mentions that CuO_x nanocomposites consist of a blend of CuO and Cu₂O, yet it fails to offer specific data concerning the ratio between the two phases, such as the atomic percentage of Cu²⁺ to Cu⁺. Is it possible to further quantify this ratio through certain measurement techniques?
16. In the transient absorption experiment, the author mentioned utilizing a pump laser power of 10 nJ. However, when describing the pump intensity in such experiments, it is more pertinent to use either the pump fluence or the number of injected photons. The author should provide one of these parameters to ascertain whether the pump power was excessively high and to evaluate the presence of multiphoton absorption effects.
17. The author mentioned that the length of Si-doped n-Al_{0.3}Ga_{0.7}N nanowire is 400 nm long, but there is no relevant data provided in the manuscript to support this. Please supplement the data accordingly.
18. There are significant issues with English grammar and formatting style, such as extra space, multiple abbreviations, reference format, incorrect spelling and grammatical structure as well as repeated word use.

Version 1:

Reviewer comments:

Reviewer #1

(Remarks to the Author)

The authors have fully addressed my comments, and the paper becomes clearer. Congratulations on the nice work, I recommend publishing this work in Nature Comm now.

Reviewer #2

(Remarks to the Author)

The authors have made the good revision. I have no other comments.

Response Letter

Manuscript Title: Multiple exciton generation boosting over 100% quantum efficiency photocatalytic photodetection

Reviewer Comments & Author Response

Reviewer #1 (Remarks to the Author):

Blue: Response to the comments

Red: Changes made to the main text and Supplementary file

The manuscript presents an interesting study on the MEG-assisted photoelectrochemical photoresponse that leads to carrier multiplication and quantum efficiency enhancement. This manuscript provides an approach to manipulating surface energy bands to extract carriers from the site of exciton generation. There are a few minor issues that need to be addressed to improve the manuscript further. Below, I outline my specific comments and suggestions.

Comments:

1. In performance evaluation, light intensity calibration is important. The authors need to provide details on how the light intensity was calibrated.

Response: Thank you for proposing this critical question regarding light intensity calibration, which is essential for ensuring the accuracy of photodetector performance evaluation. In our experiments, the light intensity was systematically calibrated using an LS-125 UV radiometer (manufactured by Shenzhen Linshang Technology) and an LH-122 solar power meter (manufactured by Shenzhen Lianhuicheng Technology). The LS-125, equipped with appropriate probes, covers the entire ultraviolet spectrum (200–400 nm), including UVA, UVB, and UVC bands, for both irradiance and energy measurements. The LH-122 is a broadband solar power meter spanning 400–1100 nm. Both instruments provide $\pm 5\%$ accuracy across the shared measurement range of $1 \mu\text{W cm}^{-2}$ to 2 W cm^{-2} , ensuring high-precision calibration under all experimental conditions.

During calibration, a real-time protocol was implemented: the light source and photodetector were fixed on an optical platform, while the probes of the light meters were positioned in close proximity to the detector surface (without blocking the light incident on the detector) to ensure

identical light exposure conditions. This setup enabled synchronous intensity monitoring throughout the experiments.

In response to your feedback, we have revised the Methods section of the manuscript to clarify this procedure: "The intensity of the light involved in the experiment were calibrated in real time by an ultraviolet radiometer (LS-125) and a solar power meter (LH-122). During the light calibration, a real-time and mature protocol was implemented."

2. The title refers to "photocatalytic photodetection," while the text uses "photoelectrochemical." Which term is more appropriate?

Response: Thanks for the suggestion. After our careful assessments, as suggested by the reviewer, the "photoelectrochemical" is, indeed, is more appropriate than "photocatalytic" in title.

According to the suggestion of the reviewers, we have revised the title of text into "Multiple exciton generation boosting over 100% quantum efficiency photoelectrochemical photodetection".

3. In Figures 3b and 3c, why does the EQE of the AlGa_N/CuO_x sample decrease with increasing light intensity? At 1000 μW/cm², the EQE is only 40%, much smaller than at 5 μW/cm², while the EQE of the AlGa_N sample does not change significantly. Also, would the EQE be even larger at lower light intensities?

Response: Thank you for the comment. In practical applications, EQE does not keep increasing indefinitely at lower light intensities. The influence of the detector's sensitivity threshold and background noise must be fully considered. When the light intensity is extremely low (e.g., < 1 μW cm⁻²), the photocurrent signal may approach the level of dark current. As a result, the signal-to-noise ratio decreases, and the actually measured EQE may fluctuate or even decline due to noise interference. Under such circumstances, it becomes difficult to accurately measure the effective EQE.

As mentioned by the comment on the EQE variation, we have included the following discussion in

the main text:

...As shown in Fig. 3c, the EQE of AlGa_N/CuO_x PD increases as the optical power intensity decreases, which is keeping with the results of responsivity. When the light intensity is 5 μW cm⁻², AlGa_N/CuO_x PD achieved an impressively high EQE of 131.5%, which was 11.1 times higher than the original PD (11.8%). Notably, due to the MEG effect triggered by the absorption of high-energy photons by CuO_x, the EQE of AlGa_N/CuO_x PD exceeds 100% without external bias. Within the range of light intensity from 5 to 1000 μW cm⁻², there is a significant change in the EQE of the AlGa_N/CuO_x PEC PD, dropping from 131.7% to 44%. In contrast, the EQE of the AlGa_N sample shows no significant change and remains at a consistently low level. The main reason for this is that, compared with the bare AlGa_N, the AlGa_N/CuO_x heterojunction sample, under the aid of MEG, generated overmany photogenerated carriers at higher light power. These excessive carrier cannot be timely transferred away, thus lowering down the EQE at high light power....

4. The authors only calculated the EQE at 255 nm. They should also calculate EQE at other wavelengths to further support the MEG effect.

Response: We are sincerely grateful for your extremely professional and perceptive comment. Your suggestion precisely aligns with the direction essential for strengthening the scientific integrity of our study. In fact, we have already calculated the EQE at multiple wavelengths. However, due to an oversight during the initial manuscript preparation, this crucial information was not adequately presented. We have now included these additional data and corresponding analyses in the revised manuscript. The relevant sections are marked for easy identification.

Supplementary Fig. 14. EQE as a function of wavelength for the bare AlGaIn and AlGaIn/CuO_x PEC PD at 0 V bias.

...Here, it needs to be mentioned that the AlGaIn/CuO_x PD there is no photocurrent can be observed under longer than 400 nm light irradiation, though the photon energy is exceeding the band gap of CuO_x. The EQE of bare AlGaIn and AlGaIn/CuO_x PEC PDs under different light wavelength (255 ~ 620 nm) was also charted in Supplementary Fig. 14. It is clearly indicated that EQE of AlGaIn/CuO_x PD exceeds 100% irradiated by ultraviolet light below 300 nm....

5. In Figure 5b, the change in Na₂SO₄ concentration of the electrolyte affects performance. As the authors mention, higher concentrations improve the conductivity of the electrolyte. However, why does a higher Na₂SO₄ concentration lead to an excess of proton concentration? Why does a higher Na₂SO₄ concentration inhibit the oxidation half-reaction at the photoanode/electrolyte interface? The authors should provide further explanation.

Response: Thank you for your insightful question. In the revised manuscript, we propose that the oxidation half-reaction at the photoanode surface follows: $2\text{H}_2\text{O} + 4\text{h}^+ \rightarrow \text{O}_2 + 4\text{H}^+$. First, sulfate ions (SO_4^{2-}) may adsorb onto the photoanode surface, competitively occupying active sites, hindering hole transfer to water molecules, and thereby lowering the oxidation kinetics. Second, the high ionic strength and viscosity of concentrated electrolytes restrict the diffusion of reaction-generated protons (H^+) away from the interface, leading to localized accumulation and transient

proton overconcentration. These combined effects, suppressing charge transfer and hindering mass transport, lead to the nonmonotonic dependence of photocurrent density on Na₂SO₄ concentration.

According to the comment, we have included the following discussion in main text:

...However, at higher ion concentrations than 0.5 M, redox at the interface is hindered, resulting in a decrease in photocurrent⁴. This phenomenon can be attributed to the adsorption of high-concentration SO₄²⁻ ions onto the photoanode surface, which occupies active sites and impedes hole transfer to reactants. Additionally, the localized accumulation of oxidation products (O₂ and H⁺) induces proton overconcentration, together suppressing oxidation kinetics.

6. Regarding the oxidation half-reaction: $\text{OH}^- + \text{h}^+ \rightarrow \text{OH}^*$, in the Na₂SO₄ solution used in the manuscript, which is neutral, there are not large quantities of OH⁻. The authors should clarify this point.

Response: We are sincerely grateful for your professional insights regarding the selection of electrolytes in our paper. Your queries have pinpointed key issues that require clarification, which are crucial for enhancing the scientific integrity of our research. Although a neutral Na₂SO₄ solution was employed, trace amounts of OH⁻ ions are present in the solution. This is due to the auto-ionization equilibrium of water, which generates OH⁻ ions. Even though the concentration of OH⁻ in a neutral solution is relatively low, the ionization equilibrium exists dynamically, continuously replenishing the OH⁻ ions involved in the reaction.

We chose Na₂SO₄ as the electrolyte primarily because it can provide a stable ionic environment with minimal interference in the PEC process. Compared to alkaline electrolytes that can supply a large amount of OH⁻ ions, Na₂SO₄ exhibits stable chemical properties under experimental conditions. This helps avoid side reactions that could potentially obscure the fundamental photoelectrochemical behavior of the system. In contrast, alkaline electrolytes may corrode the surface of nanowires during PEC testing, thereby reducing the long-term stability of PEC devices.

In the original manuscript, the oxidation reaction ($\text{OH}^- + \text{h}^+ \rightarrow \text{OH}^*$) and the reduction reaction ($\text{OH}^* + \text{e}^- \rightarrow \text{OH}^-$) proposed in our original manuscript are just the two steps of the water-splitting reaction, which finally generates H₂ and O₂. When the PEC reaction occurs in a neutral Na₂SO₄ electrolyte solution, it, ultimately, is water-splitting reaction. The oxidation reaction of water is: H₂O

$+ h^+ \rightarrow H^+ + O_2$, and finally oxygen is produced. This process can be divided into two parts. One is: $H_2O + h^+ \rightarrow H^+ + OH^*$, and the other is: $2OH^* + h^+ \rightarrow O_2 + 2H^+$. Similarly, the reduction reaction of water is $2H_2O + 2e^- \rightarrow 2OH^- + H_2$, and finally hydrogen is produced. In order to explain the PEC reaction more accurately, we have corrected the reaction formulas in the original text.

...When bare n-type AlGaIn contacts an electrolyte solution, the reduction potential difference leads to a built-in electric field, causing the surface energy band of AlGaIn to bend upward at the interface. When UV light irradiate the AlGaIn photoelectrode, photogenerated charge carriers are excited and then separated at the AlGaIn/Na₂SO₄ interface¹⁻³. The photogenerated holes migrate toward the electrolyte, reach the semiconductor/electrolyte interface, and participate in the PEC oxidation reaction ($2H_2O + 4h^+ \rightarrow O_2 + 4H^+$). The corresponding photogenerated electrons enter the counter electrode circuit through an external connection and participate in the PEC reduction reaction ($2H^+ + 2e^- \rightarrow H_2$). Thus, photogenerated electrons can be captured by electrochemical workstation according to the external circuit, resulting in a positive photocurrent. Here, the PEC device can function as a photodetector under self-powered conditions, with the assist of built-in electric field at the semiconductor/electrolyte interface driving the separation of electrons and holes. Similarly, when the bare p-type CuO_x comes into contact with the electrolyte solution, the surface energy bands of CuO_x bend downward at the interface. Upon irradiation with ultraviolet light, the photogenerated electrons migrate towards the electrolyte and participate in the reduction reaction ($2H^+ + 2e^- \rightarrow H_2$). Correspondingly, the photogenerated holes enter the counter electrode through the external circuit and take part in the oxidation reaction ($2H_2O + 4h^+ \rightarrow O_2 + 4H^+$). The migration directions of both electrons and holes are opposite to those of the previously mentioned n-type AlGaIn, so a negative photocurrent is generated.

7. The vertical axis in Figure 4b should be corrected.

Response: Thank you for the comment. We have made the correction on the vertical axis of Fig. 4b.

8. The authors should further provide SEM cross-sectional images of the nanowires before and after loading.

Response: Thank you for the valuable suggestion. Cross-sectional SEM images of the AlGa_N nanowires before and after loading are indeed critical evidence to confirm successful decoration. In the revised manuscript, we have added these images in Supplementary Figure S#, accompanied by detailed characterization data analysis.

We have revised the supplementary Fig. 2 and added introductory sentence in main text.

...As shown in Fig. 2b and Supplementary Fig. 6, scanning electron microscopy (SEM) was conducted to characterize the changes on the morphology of AlGa_N NWs before and after CBD deposition. In the top half of Fig. 2b, the morphology of the pristine AlGa_N specimen shows that the as-grown AlGa_N nanowires by MBE are vertically arranged on substrate and have a uniform morphology in size. After CBD modification, the AlGa_N nanowires are coated by copper-oxides thin layer and the CuO_x layer exhibiting the concavo-convex morphology, which is shown in the bottom half of Fig. 2b. Supplementary Fig. 6 provides the cross-sectional SEM images, where it is obvious that the CuO_x layer resides atop the AlGa_N NWs (~400 nm). Furthermore, TEM image, shown in Fig. 2c, indicates that the more deposited CuO_x mainly resides on the top of the nanowires and the side wall of nanowires are covered with thin CuO_x layer, forming a core-shell structure, which corresponds to the distribution properties of CBD deposition...

Supplementary Fig. 6. SEM cross-sectional image of nanowires. **a**, AlGaIn, **b**, AlGaIn/CuO_x.

9. The authors should provide the photocurrent response curve of Figure 6c.

Response: Thank you for your professional suggestions. The photocurrent response curve shown in Figure 6c is the core data of ultraviolet imaging and is the result of real-time monitoring during the image transmission process. In Supplementary Fig.S# of the revised manuscript, we have added this part of the data. Meanwhile, we have also provided a magnified view of the local data of the image, corresponding to the result of the line scan in the image.

More details about the imaging tests can be found in Supplementary Figure 19.

Supplementary Fig. 19. Imaging with the AlGaIn/CuO_x PEC PD. The photocurrent response curve

(a) of the mask scanned by the AlGa_xN/CuO_x PEC PD, where the enlarged view (b) of the curve within the purple rectangle corresponds to the purple line in the imaging result graph (c).

The operational principles and testing procedures of the optical imaging system, which are of great significance for understanding the entire experimental process, are clearly presented by the schematic diagram shown in Figure 6a. The mask is placed between the PEC PD and the LED light source, where it undergoes a scanning operation relying on an X-Y biaxial displacement platform. The scanning speed, which can be set within the range from 0.01 mm/s to 1 mm/s based on the desired imaging effect, and the sample interval, which can be adjusted from 0.0001 to 0.1 s, are the key factors that determine the imaging clarity. The hollow parts of the mask pattern, through which the ultraviolet light emitted by the LED can pass, serve as the main source of the photocurrent. In order to eliminate the interference of the divergent light from the LED on the quality of imaging, PVC tape with excellent light-shielding properties is used to seal the parts other than the PEC PD device (whose area is approximately 0.01 cm²). During the testing process, the PEC PD collects current data in real-time, and these data are processed by the algorithms of MATLAB software and output in the form of a heatmap, so that the final imaging results can be obtained.

10. As said by authors, the “reach-through” layer is important for extracting excitons. Is the electric field in the completely depleted CuO_x layer crucial? Please make comments on this.

Response: Thank you for the professional comment. The electric field in the completely depleted CuO_x layer is indeed of great significance. As mentioned in our manuscript, when p-type CuO_x is loaded in onto heavily n-doped AlGa_xN, due to the large difference in work function between the two, the p-type CuO_x will be completely depleted by the n-type AlGa_xN, and the energy band will bend upward at the CuO_x/electrolyte interface. The the built-in electric fields within CuO_x layer ensures that the photogenerated carrier can drift through out of CuO_x layer, without any diffusion movement, significantly accelerating carrier transport and efficiently facilitating charge separation in the entire CuO_x segment.

To response the comments from the reviewer, we have included the following discussion in main text:

...It is worthy to be noted that the formed the reach-through CuO_x layer combined with the

staggered-gap band structure of AlGa_xN/CuO_x heterojunction, shown in Fig. 1d, is perfectly fit for extracting the multiplicative electrons from CuO_x nanostructure into AlGa_xN. Once the multiple excitons were generated by high-energy photons in the CuO_x layers, an appropriate method of extracting charges is very crucial for suppressing thermal relaxation, due to the extremely short lifetime of the multiple excitons. The electric field in the “reach-through” CuO_x layer can impel the charge separation effectively. The significance of “reach-through” CuO_x layer is that photo-induced carriers can be impelled to drift across the whole depleted layer by the built-in electric fields between AlGa_xN and electrolyte, yielding much higher efficiency of charge extraction than diffusion movement. Since then, the following issue should be emphatically considered is that the characteristics of spatial distribution on optical field and the photogenerated multiple excitons....

11. To determine the existence of the MEG process and its quantum yield, TA spectroscopy probing the dynamic of ground-state bleaching is performed on a spin-coated CuO_x composite layer on quartz substrates. Here, why did the authors use different samples for testing? Whether the growth condition are consistent with the above article, whether the sample has been confirmed by other characterization methods.

Response: Thank you for your professional queries regarding the TA spectroscopy.

First, the transient absorption spectroscopy technique boasts extremely high temporal resolution, enabling it to precisely capture the rapid-changing processes of carriers. By using ultrashort - pulse lasers as both the excitation and probe light sources, the sample can be excited and probed within an extremely short time interval. This allows for real - time observation of the carriers' states and behaviors at different moments, which can be used to characterize the quantum yield of the sample.

When testing the transient absorption spectrum, we chose the CuO_x sample on a high-transparency quartz substrate instead of the Si substrate sample. This is mainly determined by the different optical and physical properties of the two substrates. The quartz substrate has significant advantages. It exhibits high transparency over a wide spectral range from ultraviolet to visible light, which is crucial for transient absorption spectroscopy measurements that require monitoring the absorption changes of the sample over a broad wavelength range. The high transparency ensures that the substrate absorbs very little light, without interfering with the sample's absorption signal, thus enabling the measured absorption spectrum to accurately reflect the sample's intrinsic

characteristics. Moreover, the quartz substrate has extremely low spontaneous fluorescence, which can reduce background interference, improve the sensitivity and accuracy of the measurement, and allow the experiment to more clearly capture the sample's transient absorption signal. In contrast, the Si substrate has obvious disadvantages. As a semiconductor material, Si has intrinsic absorption in the near - infrared to visible light region. The absorption of near - infrared light corresponding to its indirect bandgap will be superimposed on the sample's absorption signal, interfering with the analysis of the sample's absorption characteristics. Additionally, charge transfer may occur between the Si substrate and the sample, and the interface states at the interface can trap carriers, affecting the carrier dynamics process inside the sample. This makes it difficult for the measurement results to accurately reflect the sample's own carrier dynamics characteristics. In contrast, the quartz substrate has a weaker interaction with most samples, which can better ensure the sample's independence and make the measurement results more truly reflect the sample's properties.

Therefore, in this test, we used the CuO_x sample grown on a quartz substrate for the TA spectroscopy test. Regarding this point, we have made the following additional explanations in the article:

...To elucidate the underlying mechanism on the enhanced PEC performance of $\text{AlGaIn}/\text{CuO}_x$ nanowires assisted by the MEG effect, the behavior of charge generation and recombination were investigated using femtosecond transient absorption (TA) spectroscopy^{24,25}, in which low flux pulse excitation is designed to ensure that the number of excitons in the multiple photon excitation (MPE) state (i.e., absorbing multiple photons to create multiple excitons) can be neglected²⁶⁻²⁸. **In order to prevent the Si substrate from interfering with absorption spectra, TA characterization were carried out on CuO_x samples deposited on quartz substrates (see Methods and Supplementary Fig. 9).** Supplementary Fig. 10 shows the time-resolved transient absorption spectra measured at a pump energy of 4.13 eV, a ground-state bleaching (GSB) peak appears near 520 nm under the excitation pulse, corresponding to the steady-state absorption peak of CuO_x , which is consistent with the result of ultraviolet-visible absorption spectrum (Supplementary Fig. 11a)...

Second, just like the method of growing CuO_x on Si-based AlGaIn nanowires, the CuO_x on the quartz substrate was also grown using the chemical bath deposition method. Due to our oversight, only the detailed description of the former method was provided in the methods section of the

original manuscript. In the methods section of the newly submitted manuscript, we have added the detailed growth method for the latter, as follows:

...Finally, AlGaIn samples with CuO_x nanocomposite structures were obtained. Similarly, immerse the cleaned quartz substrate into Solution I. Under the same conditions, a sample of the CuO_x nanocomposite structure supported on the quartz substrate will be obtained.

Third, the CuO_x composite layer sample on the quartz substrate has also undergone XPS spectroscopy testing. This part is added to Supplementary Fig. #, which also confirms the existence of the CuO and Cu₂O composite structure. The modification is as follows:

Supplementary Fig. 9. The XPS characterization of CuO_x grown on a quartz substrate.

12. The author calculates that the external quantum efficiency of AlGaIn/CuO_x PEC PD is 131.5% , and the quantum yield of CuO_x is about 1.8 at 4.13 eV high-energy photons, both the concepts of QE and EQE are mentioned. Should emphasize the connection between the two and the numerical differences? What is the relationship between them?

Response: Thank you for your inquiry regarding quantum efficiency. We will explain the relationship and differences between QE and EQE as follows:

QE refers to the ratio of the number of excitons generated by incident photons to the number of incident photons in an optoelectronic device. It describes the device's ability to convert incident light into electrical signals and reflects the device's efficiency in absorbing and utilizing photons. It mainly depends on factors such as the material properties of the optoelectronic device, its structural

design, and the wavelength of the incident light. In the paper, the femtosecond transient absorption spectroscopy was used to measure the carrier dynamics curve, and the quantum efficiency of CuO_x on a quartz substrate was characterized to be 1.85.

EQE is the ratio of the number of carriers measured from the outside of the optoelectronic device to the number of photons incident on the device surface. It not only takes into account the device's ability to absorb and convert photons but also includes factors such as carrier collection efficiency. It is a more comprehensive indicator for evaluating the performance of optoelectronic devices. In the paper, the external quantum efficiency of the AlGa_N/CuO_x PEC PD was calculated to be 131.5%.

Both QE and EQE are used to measure the efficiency of optoelectronic devices in utilizing light. Their relationship is that EQE depends on QE to a certain extent. QE is the foundation of EQE. Only when the device effectively generates electron-hole pairs internally can a high external quantum efficiency be achieved. The difference between them is that QE focuses on the conversion efficiency of photons to electron - hole pairs inside the device, only considering the material's own ability to absorb and convert photons. In contrast, EQE considers the overall efficiency measured from the outside of the device, covering the losses in the carrier collection and transport processes in addition to the internal conversion process. Numerically, EQE is usually less than QE. However, for the two values of QE and EQE mentioned in the paper, the measurement subjects are different. The subject of QE is CuO_x, while that of EQE is the AlGa_N/CuO_x PEC PD, so there is no direct relationship between them.

Reviewer #2 (Remarks to the Author):

Blue: Response to the comments

Red: Changes made to the main text and Supplementary file

This manuscript reports an AlGa_N/CuO_x self-powered photodetector with a record-high responsivity of 270.6 mA W⁻¹ and an external quantum efficiency of 131.5% due to multiple exciton generation. While the experimental methodology is technically rigorous, several critical scientific

problems remain unclear, including the fundamental working mechanisms of the PEC photodetector and the photophysical process underlying the tunable photoresponse. Furthermore, this PEC photodetector demonstrates a relatively slower response and weaker stability compared to previously reported self-powered 2D PEC detectors. Additionally, there are significant issues with English grammar and formatting in the manuscript. Therefore, I do not believe the manuscript is suitable for publication in Nature Communications at the present stage. However, I would encourage you to resubmit a new manuscript after extensive revision.

1. The experimental evidence of p-type CuO_x and n-type AlGaN semiconductors lacks, which are key results to realize a negative and positive photocurrent signal and demonstrate the interfacial carrier transportation process.

Response: We appreciate the comment from the reviewer. This is an excellent point. The physical origin of n-type AlGaN is from the Mg doping during the MBE growth and the p-type origin is conventionally considered as the oxygen vacancy for the CuO_x . Unlike the bulk or film semiconductor, the conduction characteristic of nano-structural semiconductor cannot directly be acquired by Hall measurement. Yet the band bending from the diffusion of majority at the semiconductor/electrolyte interface can regulate the polarity of photocurrent flow, which reflects the type of semiconductors.[1] When contacting with the electrolyte solution, the surface energy band of n-type semiconductors bends upward. As a result, in Supplementary Fig. 2c, the positive photocurrent of the bare-AlGaN NW specimen implies that the AlGaN is a n-type semiconductor. For the aspect of CuO_x , the AlGaN/ CuO_x heterostructure, indeed, could blur the direct evidence of the surface band bending. In order to get rid of the interfering from the heterojunction, the specific photoelectrode, with depositing ultra-thick CuO_x layer on FTO substrate, was prepared for i-t test to observe the polarity of the photocurrent aroused by the bare CuO_x . As shown in Supplementary Fig. 2c, the photocurrent with the negative polarity means the CuO_x as the p-type semiconductor.

As mentioned this point by the review, to facilitate readers' better understanding, in the revised manuscript, we added experiment on photocurrent measurements of bare CuO_x . Also, we included the following discussion in the main text.

...Herein, the n-AlGaN nanowires were treated with surface modulation by depositing CuO_x

composite, composed of CuO and Cu₂O nano-mixture. Regarding to conduction type of CuO_x, the AlGaIn/CuO_x heterostructure, indeed, could blur the direct evidence of the surface band bending. In order to get rid of the interfering from the heterojunction, the specific photoelectrode, with depositing CuO_x layer on FTO substrate, was prepared for *I-t* test to observe the polarity of the photocurrent aroused by the bare CuO_x. When the bare p-type semiconductor contacts the electrolyte, the energy band at the semiconductor/electrolyte interface bends downward (Supplementary Fig. 2b), resulting in a negative photocurrent under ultraviolet light illumination. As shown in Supplementary Fig. 2c, the photocurrent with the negative polarity means the CuO_x as the p-type semiconductor. According to our best knowledge, the operating mechanisms on the manipulation of carrier dynamics for the n-type III nitrides decorated by the CuO_x could be analyzed through two types of role definition of the CuO_x in the PEC process: as a surface co-catalyst to improve surface chemical dynamics² or as the p-type light absorbing layer to construct the pn junction for inner structure optimization^{3,4}...

Supplementary Fig. 2. The energy band structures of the n-AlGaIn/electrolyte (a) and p-CuO_x/electrolyte (b) hetero-interface. c, The current signal of n-AlGaIn and p-CuO_x PEC PDs.

When bare n-type AlGaIn contacts an electrolyte solution, the reduction potential difference leads to a built-in electric field, causing the surface energy band of AlGaIn to bend upward at the interface. When UV light irradiate the AlGaIn photoelectrode, photogenerated charge carriers are excited and then separated at the AlGaIn/Na₂SO₄ interface¹⁻³. The photogenerated holes migrate toward the electrolyte, reach the semiconductor/electrolyte interface, and participate in the PEC oxidation reaction ($2\text{H}_2\text{O} + 4\text{h}^+ \rightarrow \text{O}_2 + 4\text{H}^+$). The corresponding photogenerated electrons enter the counter electrode circuit through an external connection and participate in the PEC reduction reaction ($2\text{H}^+ + 2\text{e}^- \rightarrow \text{H}_2$). Thus, photogenerated electrons can be captured by electrochemical workstation

according to the external circuit, resulting in a positive photocurrent. Here, the PEC device can function as a photodetector under self-powered conditions, with the assist of built-in electric field at the semiconductor/electrolyte interface driving the separation of electrons and holes. Similarly, when the bare p-type CuO_x contacts with the electrolyte solution, the surface energy bands of CuO_x bend downward at the interface. Upon irradiation with ultraviolet light, the photogenerated electrons migrate towards the electrolyte and participate in the reduction reaction ($2\text{H}^+ + 2\text{e}^- \rightarrow \text{H}_2$). Correspondingly, the photogenerated holes enter the counter electrode through the external circuit and take part in the oxidation reaction ($2\text{H}_2\text{O} + 4\text{h}^+ \rightarrow \text{O}_2 + 4\text{H}^+$). The migration directions of both electrons and holes are opposite to those of the previously mentioned n-type AlGaIn, so a negative photocurrent is generated.

Reference:

[1] Fang, S. *et al.* Light-Induced Bipolar Photoresponse with Amplified Photocurrents in an Electrolyte-Assisted Bipolar p–n Junction. *Advanced Materials* **35**, 2300911 (2023).

2. AlGaIn/CuO_x-nanowire is grown on n-Si (111) substrate, forming Si/AlGaIn/CuO_x heterostructure. Certainly, the multi-interfacial electron transfer process also plays significant role in improving photodetection performance. However, the influence of the substrate and interface carrier transfer on this photodetector has yet to be explored and studied.

Response: Thanks the reviewer for the comment on the key point. During the period of the preparing the initial manuscript, we did think about the influence of the Si substrate/AlGaIn interface on the carrier transport. We appreciate the comment from the reviewer. This is an excellent point. As described in main text, n-AlGaIn nanowires were vertically grown on n-Si (111) substrate. Specifically, to achieve good electroconductibility between the n-AlGaIn and Si, the heavily n-type doped Si substrates were adopted for MBE growth. On the one hand, as the results shown by the FDTD simulation, most of UV light could be absorbed by the AlGaIn/CuO_x-nanowire, which determines that a great many of the carriers generate within nanowires, rather than at the AlGaIn/Si interface or in the Si substrate. On the other hand, because of the relatively small band offset between the n-Si and n-AlGaIn, as shown in Supplementary Fig. 2c, either the electrons or holes

separated from CuO_x/AlGa_N interface can easily migrate across the AlGa_N/Si interface and into Si substrate. This means that, in CuO_x/AlGa_N/Si multi-heterojunctions, the carrier transfer behavior is dominated by the CuO_x/AlGa_N interface.

As mentioned the influence of the substrate and interface carrier transfer on this photodetector, we have included the following discussion in the main text:

...Similarly, when the p-CuO_x/n-AlGa_N junction is exposed under λ_2 light, a small magnitude of positive photocurrent generates in the circuit loop for this case, which is opposite to the direction of photocurrent for λ_1 light in case of Fig. 1b. In the other word, detecting the polarity switching of photocurrent between high- and lower- energy light is an effective method to confirm the truth of direction change of bend band at p-CuO_x/electrolyte interface, as the deposited amount of CuO_x increase on the AlGa_N nanowire. **It is worthy to note that the photoelectrode with configuration of CuO_x/AlGa_N/Si multi-heterojunction was designed for optimizing band alignment and carrier transfer. Generally speaking, as the results shown by the FDTD simulation (see below), most of UV light could be absorbed by the AlGa_N/CuO_x-nanowire, which determines that a great many of the carriers generate within nanowires. Thus, for the upper CuO_x/AlGa_N interface, it was assigned to photogenerated carrier regulation. For the lower AlGa_N/Si interface, the heavily n-type doped Si substrates were adopted for plasma-enhanced molecular beam epitaxy (MBE) growth to achieve good electroconductibility between the n-AlGa_N and Si. As shown in Supplementary Fig. 2c, due to the characteristics of the n-AlGa_N/n-Si band alignment, either the electrons or holes separated from CuO_x/AlGa_N interface can easily migrate across the AlGa_N/Si interface and into Si substrate. It means that the AlGa_N/Si interface plays the role as the linker for electric conduction.**

3. The author pointed out that the working principle of the PEC photodetector is driven by an oxidation reaction ($\text{OH}^- + \text{h}^+ \rightarrow \text{OH}^*$) and a reduction reaction ($\text{OH}^* + \text{e}^- \rightarrow \text{OH}^-$). However, it is unclear how the author confirmed this specific chemical reaction. Previous reports have indicated that the water redox reaction, which generates H₂ and O₂, also contributes to photocurrent generation.

Response: Thank the reviewer a lot for pointing out the inaccuracy. As suggested by reviewer, the

basic working principle of the PEC photodetector, indeed, is the water redox reaction, producing H₂ and O₂ and contributing to the generation of photocurrent. In the original manuscript, the oxidation reaction ($\text{OH}^- + \text{h}^+ \rightarrow \text{OH}^*$) and the reduction reaction ($\text{OH}^* + \text{e}^- \rightarrow \text{OH}^-$) are just the two steps of the water-splitting reaction, which finally generates H₂ and O₂. When the PEC reaction occurs in a neutral Na₂SO₄ electrolyte solution, ultimately, it is a water-splitting reaction. The oxidation reaction of water is: $\text{H}_2\text{O} + \text{h}^+ \rightarrow \text{H}^+ + \text{O}_2$, and finally oxygen is produced. This process can be divided into two parts. One is: $\text{H}_2\text{O} + \text{h}^+ \rightarrow \text{H}^+ + \text{OH}^*$, and the other is: $2\text{OH}^* + \text{h}^+ \rightarrow \text{O}_2 + 2\text{H}^+$. Similarly, the reduction reaction of water is $2\text{H}_2\text{O} + 2\text{e}^- \rightarrow 2\text{OH}^- + \text{H}_2$, and finally hydrogen is produced. In order to explain the PEC reaction more accurately, we have corrected the reaction formulas in the original text.

According to the suggestion of the reviewer, we have made the revision in Supplementary Fig. 2 as follow.

When bare n-type AlGaIn contacts an electrolyte solution, the reduction potential difference leads to a built-in electric field, causing the surface energy band of AlGaIn to bend upward at the interface. When UV light irradiate the AlGaIn photoelectrode, photogenerated charge carriers are excited and then separated at the AlGaIn/Na₂SO₄ interface¹⁻³. The photogenerated holes migrate toward the electrolyte, reach the semiconductor/electrolyte interface, and participate in the PEC oxidation reaction ($2\text{H}_2\text{O} + 4\text{h}^+ \rightarrow \text{O}_2 + 4\text{H}^+$). The corresponding photogenerated electrons enter the counter electrode circuit through an external connection and participate in the PEC reduction reaction ($2\text{H}^+ + 2\text{e}^- \rightarrow \text{H}_2$). Thus, photogenerated electrons can be captured by electrochemical workstation according to the external circuit, resulting in a positive photocurrent. Here, the PEC device can function as a photodetector under self-powered conditions, with the assist of built-in electric field at the semiconductor/electrolyte interface driving the separation of electrons and holes. Similarly, when the bare p-type CuO_x comes into contact with the electrolyte solution, the surface energy bands of CuO_x bend downward at the interface. Upon irradiation with ultraviolet light, the photogenerated electrons migrate towards the electrolyte and participate in the reduction reaction ($2\text{H}^+ + 2\text{e}^- \rightarrow \text{H}_2$). Correspondingly, the photogenerated holes enter the counter electrode through the external circuit and take part in the oxidation reaction ($2\text{H}_2\text{O} + 4\text{h}^+ \rightarrow \text{O}_2 + 4\text{H}^+$). The migration directions of both electrons and holes are opposite to those of the previously mentioned n-type

AlGa_N, so a negative photocurrent is generated.

4. Since OH⁻ can consume the photogenerated holes, why did not select alkaline electrolyte in the PEC measurement to further improve photoresponse?

Response: Thank you very much for the professional question. We selected Na₂SO₄ as the electrolyte mainly because it can create a stable ionic environment and causes minimal interfering with the PEC process. Compared with alkaline electrolytes that can provide a large quantity of OH⁻ ions, Na₂SO₄ features stable chemical properties under experimental conditions. This enable to avoid side reactions that may obscure the fundamental PEC behavior of the system. Alkaline electrolytes might even corrode the surface of nanowires during PEC testing, thereby reducing the long-term stability of PEC devices. We have attached the long-term stability test of the AlGa_N/CuO_x PEC PD in an alkaline NaOH electrolyte solution. After 12000 seconds, the photoresponse performance drops over 50%, indicating that this detector is not suitable for long-term operation in alkaline electrolyte.

According to the comment of reviewer, we have included the following discussion about the electrolyte selection and also attached the long-term photocurrent test results of the AlGa_N/CuO_x PEC PD in alkaline electrolyte, as shown in Supplementary Fig. 18.

...And more notably, the main reason of selecting Na₂SO₄ electrolyte is that it could create relatively stable ionic environment, suppressing interfering with the photoelectrochemical (PEC) process by the side reactions. In addition, though the alkaline electrolyte could provide higher OH⁻ ion concentration and might improve the photoresponse, the original Na₂SO₄ is not suitable to be replaced with the alkaline electrolyte, in consideration of chemical stability of PEC PD. Alkaline electrolytes might corrode the surface of nanowires during PEC testing, thereby reducing the long-term stability of PEC devices. As shown in Supplementary Fig. 18, the the photoresponse performance drops over 50% within initial 12000 s....

Supplementary Fig. 18. 12000-second *I-t* test of AlGaIn/CuO_x PEC PD in alkaline NaOH electrolyte solution (0.1 M) at 255 nm light irradiation (1 mW cm⁻²).

5. Why the light intensity-photocurrent is nonlinear trend in Figure 3b? However, this nonlinear response is disadvantage for practical application of photodetector. Please comment on whether such kind of photodetector could become commercialized?

Response: Thank you for the professional comment. As the said by the reviewer, the nonlinear response, indeed, is disadvantage for practical application of photodetector. The nonlinear response photocurrent in Fig. 3b is mainly ascribed to the nonlinear coordinate of horizontal axis.

To void the misunderstanding from the readers, we have altered the coordinate of horizontal axis Fig. 3b in main text, as follow:

(Fig. 3b)

Thank you, as well, for your question regarding commercial viability. Currently, our detectors have demonstrated high responsivity at the laboratory stage. Regarding to the issues about the commercialization of PEC photodetectors, it is a very forward-looking question. We have added the comment on the commercialized PEC PD in main text:

...Because of the outstanding photoresponse performance, it is feasible to make discussion on the proposed PEC PD for further application and even for the commercialization in future. Totally, there are still several issues to be solved, such as device packing and system integration. At present, from the prospective of the authors, two technical issues that need to be addressed. Because of no external supply, the photoresponse of the self-powered PEC PD should be improved further. Certainly, the MEG involved PEC PD in this paper provides one effective solution. For another thing, the liquid electrolyte as well as the bulky vessel impedes the commercial progress. In facts, the traditional water and electrolytic vessel for the PEC PD could be replaced by the solid electrolyte based capsule, which significantly reduces the volume of PEC PD, less than several-hundredths of original PEC cells. It greatly improves the practicality in the field of portable and wearable electronics^{38,39}....

6. Why does the current in Figure 5g suddenly surge during the moment of illumination, while that of Figure 4(a) and 5(a) does not?

Response: Thank you for proposing this question. In Figure 5g, the current surges sharply at the moment of illumination, which is manifested as a significant spike in the photocurrent. In contrast, the spikes in Figures 5a and 5b are relatively less obvious in terms of visual effects. This is mainly caused by two factors. Firstly, there are differences in the data visualization process. In Figures 5a

and 5b, multiple sets of $i-t$ curves are plotted on the same graph, resulting in the curves being visually compressed. Secondly, the ratios of the horizontal and vertical coordinates in Figures 5a and 5b are inconsistent with those in Figure 5g, and this difference further weakens the visual effect of the spikes in Figures 5a and 5b.

Nevertheless, when Figures 5a and 5b are magnified, or the ratios of the horizontal and vertical coordinates are adjusted (as shown in the following figures), it can be clearly observed that the current in Figures 5a and 5b also surges sharply at the moment of illumination.

7. In Fig 3d, the Nyquist plot should be fitted with equivalent circuit model to better understand the carrier dynamics.

Response: Thank you for your suggestion. Fitting the Nyquist plot into an equivalent circuit model can indeed provide a better understanding of the carrier dynamics. We have already added equivalent circuit model to the manuscript.

...To evaluate the contribution of CuO_x nanocomposite in PEC process, electrochemical impedance spectroscopy (EIS) was measured under illumination of 255 nm light. EIS diagram could mirror the properties at the semiconductor/electrolyte interface. The diameter of EIS curve reveals the interfacial charge transfer resistance (R_{ct}), providing information about the charge transfer kinetics at the interface. The EIS plots of the bare AlGaN and the CuO_x decorated AlGaN samples are demonstrated in Fig. 3d. In order to quantitatively study the change of the R_{ct} , we employed an equivalent circuit that consists of a series resistance (R_s), a bulk resistance (R_1), a constant-phase element (CPE), and R_{ct} , which is shown in the inset of Figure 3d. The value of R_{ct} , which decreased by more than an order of magnitude, dropped from $3.77 \times 10^6 \Omega \text{ cm}^2$ for AlGaN to $2.44 \times 10^5 \Omega \text{ cm}^2$ for AlGaN/CuO_x. The AlGaN/CuO_x sample shows the smaller R_{ct} among these samples, which implies that the CuO_x deposition can facilitate ultra-fast charge transfer at the interface, benefiting the exciton extraction from CuO_x layer...

8. The 3000-second photocurrent test appears to inadequately reflect the stability performance of the PEC photodetector. Therefore, it is recommended that the photocurrent measurement time should exceed 30000 seconds or longer.

Response: Thank you for your suggestion. We have updated the results of the stability test in the

manuscript. A photocurrent test was conducted over a duration of 50,000 seconds, and the results show no obvious attenuation in photoresponse.

(Fig. 5g)

Generally speaking, the durability and long-term stability of multi-cycle photodetectors are the crucial characteristics for the further application. Fig. 5g shows the long-term on/off response of AlGaIn/CuO_x samples exposed to the air for one year under illumination of 200 μW cm⁻² without bias. The photocurrent of AlGaIn/CuO_x samples only exhibits slight attenuation after 50,000 seconds of long-term operation, highlighting the remarkable stability of AlGaIn/CuO_x samples in multiple cycles. Actually, because of the surface passivation by deposited CuO_x, the AlGaIn nanowires were substantially mitigated from the photo-induced corrosion, resulting in excellent durability and stability for AlGaIn/CuO_x PD.

9. The specific detectivity and -3 dB cutoff frequency are also another important parameter for evaluating the device performance and should be calculated.

Response: Thank you for your suggestions. We have added the specific detectivity and -3 dB cutoff frequency in the revised manuscript, and the specific changes have been marked in red in the revised manuscript.

Supplementary Fig. 8. The specific detectivity of bare AlGaN and AlGaN/CuO_x PEC PDs under 255 nm illumination at different light power.

...Besides of the responsivity and EQE, specific detectivity (D^*) is another crucial parameter for assessing the ability of PD to detect weak signals. When the noise is mainly dominated by the shot noise of the dark current (I_{dark}), the equation of the specific detectivity can be simplified as:

$$D^* = \frac{R\sqrt{S}}{\sqrt{2qI_{dark}}} \quad (3)$$

Where R is the responsivity of PD, S is the area and q is the electron charge. The calculated D^* values for different light intensities are shown in Supplementary Fig. 8. Impressively, the AlGaN and AlGaN/CuO_x PEC PDs exhibit remarkable D^* value of 5.56×10^{11} and 6.17×10^{12} Jones at $5 \mu\text{W cm}^{-2}$, respectively, revealing the ability of AlGaN/CuO_x PEC-type PD to detect weak light...

(Fig. 5e)

Supplementary Fig. 17. The normalized response curve at transmitting frequencies of 1, 10, 100, 500, 1k, 2k, 5k, 10k and 20k Hz, respectively.

...It is not difficult to see that the improved response speed of AlGaIn/CuO_x PD is most likely due to the accelerated charge transfer process and redox rate at the interface. To thoroughly study the frequency characteristic of the AlGaIn/CuO_x PEC PD in electrolyte, frequency-based tests were conducted, with presenting the time response of the PEC PD in the 1 - 20k Hz frequency range in Supplementary Fig. 17. It can be observed that the output waveform remains stable as a square wave with clear rising/falling edges up to 1 kHz, yet when the frequency increases to 10 kHz, the output waveform transitions from a square wave to a triangular wave, probably because the PD reaches its response-speed limit and lacks sufficient time to stabilize at high frequencies. The frequency-response curve (Fig. 5e) was obtained by measuring the amplitudes of output signals at different frequencies, enabling to estimate that the -3 dB cut-off frequency of AlGaIn/CuO_x PEC PD in sodium sulfate solution is higher than 10 kHz....

10. The response time of AlGaN/CuO_x PEC photodetector is in millisecond scale, which is far slower than previous reported microsecond-level photodetector. The measured result may be incorrect and influenced by machine response time of electrochemical workstation. It is suggested that the oscilloscope is used to collect the signal in the frequency range of 1–100 kHz.

Response: We sincerely appreciate the professional comment. In accordance with the suggestions from the reviewer, we have carefully rechecked the optical path, the modulator for controlling the light source, and the electrochemical workstation. After making the adjustments on the switching speed of light source modulator, we have archived significant improvement in the response time of the PEC photodetector. The updated data have been incorporated into the manuscript and marked in red for easy identification.

...To display the light response speed of the PD, Fig. 5d (AlGaN/CuO_x) and Supplementary Fig. 16 (AlGaN) show the response time (t_r) and decay time (t_d) intervals of the photocurrent curve at a bias potential of 0 V under 255 nm illumination. The t_r and t_d are defined as intervals from 10% to 90% of the maximum photoelectric values and from 90% to 10%, respectively. The corresponding t_r and t_d of AlGaN/CuO_x are 183.3 μ s and 154.6 μ s, respectively, while the response speeds of the bare AlGaN nanowires was 360.8 μ s and 241.3 μ s, respectively. It is not difficult to see that the improved response speed of AlGaN/CuO_x PD is most likely due to the accelerated charge transfer process and redox rate at the interface....

Besides, as suggested by the review, we have collected response signal in the frequency range of 1 ~ 20k Hz, shown in Supplementary Fig. 17. However, the collected signal become severely distorted at higher frequency, which induces the failure at 100 kHz. In order to improve the frequency characteristic in future, we have added the comments in main text as follow:

...In our opinions, although achieving ultra-high frequency response for PEC devieces is still a challenge, a few promising strategies were proposed, such as interface engineering to eliminate recombination, nanostructure design to reduce the carrier diffusion path, circuit optimization for impedance matching....

Supplementary Fig. 17. The normalized response curve at transmitting frequencies of 1, 10, 100, 500, 1k, 2k, 5k, 10k and 20k Hz, respectively.

11. In the FDTD simulation, AlGa_N and CuO are modeled with rectangular structures, whereas their actual morphology is that of core-shell nanowires. Might this simplification impact the accuracy of the simulation results?

Response: We sincerely appreciate your professional comment. As said by the reviewer, the AlGa_N-CuO_x nanowires, indeed, have the core-shell morphology. Since the CuO_x thickness on the sidewall of NW is much thinner than on the top, we thought of the CuO_x Capping layer as the dominant factor in manipulating light incidence and propagation. Thus we simplified the AlGa_N-CuO_x nanowire modeling during the first FDTD simulation. Here, to effectively response the demands from the review, we have updated the process of FDTD simulation, replacing the simple rectangular structures with core-shell models. (The thickness of CuO_x was set as 3 nm on sidewall. And the setting value of thickness on top are consistent with the previous simulation.) As shown in Fig. xx, there are no obvious variation of the photon energy dependence of distribution of electric field in

the nanowires.

To address this issue, we have updated the Fig. 4 e-j in main text as well as the Supplementary Fig. 15.

Supplementary Fig. 15. Schematic diagram of the numerical model of AlGaIn/CuO_x nanowires heterojunction.

12. It is recommended that the authors add the application potential of the research (e.g., UV imaging) at the end of the abstract to highlight the practical significance.

Response: Thank you very much for your reasonable suggestions on revising the abstract of our paper. The revisions made are as follows:

Inspiringly, we achieve MEG effect to be applied for benefiting photoresponse enhancement, which impressively induces an external quantum efficiency of 131.5% and record-high responsivity of 270.6 mA W⁻¹ at 255 nm for the CuO_x/AlGa_{0.5}N PEC PDs. **Owing to the superior photoresponse, the potential of this proposed photodetector can be extended for applications in weak-light UV imaging.** This work, which verifies the validity of MEG in PEC application, pioneers a breakthrough solution for designing self-powered PEC based optoelectronics.

13. The author employed transient absorption technology in the manuscript to measure the carrier dynamics at corresponding locations in the CuO_x and fitted the data using a biexponential function. In practice, when fitting carrier dynamics curves, the influence of instrumental response on the experiment generally needs to be considered.

Response: We sincerely appreciate the professional comment. As said by the reviewer, in ultrafast laser measurement, the finite time resolution of the instrument (such as laser pulse width, detector delay) will cause the convolution effect between the measured curve and the actual physical process. If not corrected, the fitting results (such as lifetime values) may be biased. The instrument response function (IRF) of the laser pulse is measured and deconvolution is performed to improve the fitting accuracy. If the instrument response needs to be considered and the instrument response function $IRF(t)$ is known, assuming the carrier dynamics follows a biexponential function ($I(t) = A_1 e^{-t/\tau_1} + A_2 e^{-t/\tau_2}$), where A_1 and A_2 are the amplitudes of the two exponential terms respectively, and τ_1 and τ_2 are the time constants related to the two carrier lifetimes respectively. We convolve the theoretical model with the response function to obtain the fitting function ($I_{fit}(t) = I(t) \otimes IRF(t)$), and then optimize the parameters (A_1 , A_2 , τ_1 and τ_2) to make the convolved curve fit the experimental data better.

However, in our transient absorption characterization, the instrument response time is 35 fs, while the carrier lifetime is much longer than 1 ps. Since the instrument response time is far shorter

than the time of the carrier dynamics process, the instrument response can be neglected, and it is feasible to directly use the bi-exponential function to fit the data. There are also similar cases in previous works:

Study I: *Internal quantum efficiency higher than 100% achieved by combining doping and quantum effects for photocatalytic overall water splitting* (<https://doi.org/10.1038/s41560-023-01242-7>)

They probed the dynamics of ground state bleaching through transient absorption (TA) spectroscopy with different pump energies under low pump intensity to evaluate the MEG process of CdTe-4.2/V-In₂S₃-3 and its yield.

Study II: *Dye-Sensitized Multiple Exciton Generation in Lead Sulfide Quantum Dots* (<https://doi.org/10.1021/jacs.2c07109>)

Kinetics at the first exciton bleach minimum of PbS/Py with different pump pulse energy after normalization at tails. The solid curves are the fittings to the data points used to extract ΔA at different delay time.

Study III: *Multiple exciton generation in tin-lead halide perovskite nanocrystals for photocurrent quantum efficiency enhancement* (<https://doi.org/10.1038/s41566-022-01006-x>)

GSB dynamics normalized at the long decay tail under different pump photon energies. The solid lines are exponential decay fittings.

As mentioned this point by the review, to facilitate readers' better understanding, in the revised manuscript, we have made revisions in the main text:

...To determine the existence of MEG process and its quantum yield, TA spectroscopy probing the dynamic of ground-state bleaching is performed on spin-coated CuO_x composite layer on quartz substrates. Since the TA instrumental response time (35 fs) is much shorter than the carrier lifetime (~ 1 ps), the influence of the instrument response is negligible, and it is feasible to the fit carrier dynamics curves using a biexponential function. Under small pump energy of $h\nu < 2E_g$ at low pump intensity, the GSB decay of CuO_x only single exponential decay with a lifetime of > 10 ns, indicating the neutral single-exciton recombination....

14. Could you elaborate on the reasoning behind choosing LED lamps over lasers, which possess superior monochrome and directional properties, for use in the optical imaging experimental device? Furthermore, it is necessary to provide details regarding the processing method for optical imaging data, including the software and algorithms used, as well as the specifications for the size of the hollow part of the mask, the size of the laser spot, and the scanning speed within the optical imaging system.

Response: Thank you for the comment regarding the selection of the light source. First, both LEDs and lasers can serve as light sources in this experiment. However, we chose an LED over a laser for the following reasons:

- (1) The large emission area of the LED can provide uniform illumination across the sample area. Laser speckle noise can easily introduce artifacts, which is crucial for the quantitative imaging of nanowire arrays.
- (2) The lifespan of LEDs, conventionally, is longer than the LDs, which is conducive to long-term stability test for PEC PD.

(2) LED can be directly modulated by current. In contrast, lasers require external modulators, which not only increase the complexity of the system and signal delay but also raise costs. LEDs are much more cost-effective compared to lasers.

(3) LED light sources comply with the IEC 60825 Class 1 safety standard. They can be used safely without additional protective measures. Their low coherence and non-focused characteristics significantly reduce the risk of human exposure, making them particularly suitable for long-term biological imaging applications.

Taking all these factors into account, we selected LEDs as the light source.

To address this issue, we have added elaboration on the choosing LED source into the Method section of main text:

...Apart from taking into account the convenience and biosecurity of experimental operation, the long lifespan of LED light source is conducive to long-term stability characterization....

In addition, the introduction of detailed information regarding the data processing method for optical imaging has been added to Supplementary Figure 19 in the supporting document.

Supplementary Fig. 19. Imaging with the AlGaIn/CuO_x PEC PD. The photocurrent response curve (a) of the mask scanned by the AlGaIn/CuO_x PEC PD, where the enlarged view (b) of the curve within the purple rectangle corresponds to the purple line in the imaging result graph (c).

The operational principles and testing procedures of the optical imaging system, which are of great significance for understanding the entire experimental process, are clearly presented by the schematic diagram shown in Figure 6a. The mask is placed between the PEC PD and the LED light source, where it undergoes a scanning operation relying on an X-Y biaxial displacement platform. The scanning speed, which can be set within the range from 0.01 mm/s to 1 mm/s based on the desired imaging effect, and the sampling interval, which can be adjusted from 0.0001 to 0.1 s, are the key factors that determine the imaging clarity. The hollow parts of the mask pattern, through which the ultraviolet light emitted by the LED can pass, serve as the main source of the photocurrent. In order to eliminate the interference of the divergent light from the LED on the quality of imaging, PVC tape with excellent light-shielding properties is used to seal the parts other than the PEC PD device (whose area is approximately 0.01 cm²). During the testing process, the PEC PD collects current data in real-time, and these data are processed by the algorithms of MATLAB software and output in the form of a heatmap, so that the final imaging results can be obtained.

15. The article mentions that CuO_x nanocomposites consist of a blend of CuO and Cu₂O, yet it fails to offer specific data concerning the ratio between the two phases, such as the atomic percentage of Cu²⁺ to Cu⁺. Is it possible to further quantify this ratio through certain measurement techniques?

Response: Thank you for the comment on the point regarding the quantification of Cu⁺ and Cu²⁺ ratios in the CuO_x nanocomposite. We acknowledge that specifying the phase composition is essential for understanding its electronic and catalytic properties. In response to your suggestion, we employed X-ray photoelectron spectroscopy (XPS) to quantify the atomic percentages of Cu²⁺ and Cu⁺, and supplemented the revised manuscript with the following analyses:

The principle of XPS is based on the photoelectric effect. When the sample is irradiated with a specific-energy x-ray beam, the inner-shell electrons of the atoms in the sample absorb the energy of the x-rays and escape from the atoms as photoelectrons after overcoming the atomic binding energy. These photoelectrons carry information about the element types and chemical states in the sample. For copper oxides, copper atoms in different oxidation states (such as Cu⁺ and Cu²⁺) have different electron-cloud distributions around them, resulting in differences in the binding energies of their inner-shell electrons. By measuring the kinetic energy of these photoelectrons and

combining it with the known x-ray energy, we can calculate the binding energy of the photoelectrons and thus determine the oxidation state of copper.

When processing the XPS data, we first performed background subtraction on the original Cu2p spectrum. In actual measurements, in addition to the photoelectron signals generated by the sample, there are also some background signals, such as inelastically scattered electrons. After background subtraction, we used a Gaussian-Lorentzian mixed function to fit the peaks of the Cu2p spectrum. This is because the XPS peaks measured experimentally are usually a mixture of Gaussian and Lorentzian distributions, and using the Gaussian-Lorentzian mixed function can more accurately fit the peaks corresponding to different oxidation states of copper.

For Cu^+ and Cu^{2+} , they correspond to different characteristic peaks in the Cu2p spectrum. Generally, the characteristic peak of Cu^{2+} has a relatively high binding energy, while that of Cu^+ has a relatively low binding energy. Through precise peak fitting, we can determine the position, area, and other parameters of these two peaks. Since the peak area is proportional to the number of copper atoms in the corresponding oxidation state, we can calculate the atomic percentages of Cu^{2+} and Cu^+ based on the ratio of the peak areas. After careful analysis and calculation, we determined that the ratio of atomic percentage of Cu^{2+} to Cu^+ is one to one.

According to the comment of review, we have now added this data and the corresponding analysis in the revised manuscript. We have also supplemented the detailed process of XPS measurement and data processing to ensure the reliability and reproducibility of the results.

...To elucidate the chemical properties and electronic states of the decorated AlGaN NWs, X-ray photoelectron spectroscopy (XPS) measurements were performed. In Fig. 2a, the XPS spectra of the CBD-modified AlGaN NWs shows typical peaks of $\text{Cu}2p_{3/2}$ and $\text{Cu}2p_{1/2}$, representing features of Cu_2O and CuO , respectively^{22,23}. Two binding energy peaks (green curve) in Fig. 2a at 933.2 and 953.1 eV are associated with Cu_2O , while the other two smaller binding energy peaks (orange curve) at 934.6 and 954.5 eV are attributed to CuO . Additionally, the satellite peaks of $\text{Cu}2p_{3/2}$ and $\text{Cu}2p_{1/2}$ also confirm the existence of CuO on AlGaN. **Since the peak area shows proportionality to the number of copper atoms in their corresponding oxidation states, by calculating the ratio of peak areas, it is evident that the combined atomic ratio of Cu^{2+} to Cu^+ are nearly one to one (1:1).** On the one hand, the peak at 529.5 eV corresponds to the metal-oxygen bonding; on the other hand,

the peak locating at 531.6 eV could be attributed to surface hydroxide in the XPS spectrum of O1s (see Supplementary Fig. 5)²². All of these XPS results demonstrate that the CuO_x are well-structured on AlGaIn nanowires by CBD method....

16. In the transient absorption experiment, the author mentioned utilizing a pump laser power of 10 nJ. However, when describing the pump intensity in such experiments, it is more pertinent to use either the pump fluence or the number of injected photons. The author should provide one of these parameters to ascertain whether the pump power was excessively high and to evaluate the presence of multiphoton absorption effects.

Response: Thank you for the very professional comment. The pump power 10 nJ mentioned in our manuscript actually represents the pump fluence, with unit typically being J cm⁻². We have made the corresponding amendments in the revised manuscript.

...The pump is at 300 nm with a pulse fluence of 10 nJ cm⁻²....

Furthermore, the comment from the reviewer on the multiphoton absorption effects is extremely professional. Thanks again! We excluded the multiphoton absorption events via calculating the exciton population ratio of excitons generated directly following excitation and long delays when Auger recombination is complete (R_{pop}) at low pump fluence.

According to the comment of the review, we have made the revision in main text as well as the in Supplementary as follow:

...Where A and B are the absolute signal intensity of before and after Auger recombination. Here, as shown in Supplementary Fig. 13, it is worthy to note that the multiphoton absorption events are excluded to ascertain the behavior of the photon absorption and further support the MEG effects.

Supplementary Fig. 13. The ground state bleaching dynamics of CuO_x under different pump fluence.

(a) GSB dynamics and R_{pop} under different pump fluence at 300 nm.

The TA spectroscopy was conducted for probing the dynamics of ground state bleaching (GSB). The average number of excitons generated per absorbed photon (QY) can be determined by following equation:

$$R_{pop} = \frac{J_0 \times \sigma \times QY}{1 - e^{-J_0 \times \sigma}}$$

Where R_{pop} is the exciton population ratio of excitons generated directly following excitation and long delays when Auger recombination is complete, J_0 is the photo fluence of the pump pulse, σ is the absorption cross section at the pump wavelength, and at low fluence where $(1 - e^{-J_0 \times \sigma}) \rightarrow J_0 \sigma$, then $R_{pop} \rightarrow QY$. The GSB dynamics of CuO_x at different fluence generated by 300 nm light are shown in Supplementary Fig. Xxa. The R_{pop} at 7, 10 and 13 nJ cm⁻² are almost constant. R_{pop} reaches fluence independent regime, indicating the absence of multiphoton absorption.

17. The author mentioned that the length of Si-doped n-Al_{0.3}Ga_{0.7}N nanowire is 400 nm long, but there is no relevant data provided in the manuscript to support this. Please supplement the data accordingly.

Response: Thank you for your query regarding the length of the nanowires. It's true that the original manuscript lacked evidence to support the claim that the length of the n-Al_{0.3}Ga_{0.7}N nanowires is 400 nm. In the revised manuscript, we have added SEM images to confirm this information. As shown in Supplementary Fig. 6, we have provided SEM images of the cross-sections of the AlGa_{0.3}N

nanowires both before and after loading the CuO_x composite structure, clearly showing the length of the nanowires. We also have made revision in main text.

Supplementary Fig. 6. SEM cross-sectional image of nanowires. **a**, AlGaIn, **b**, AlGaIn/ CuO_x

...As shown in Fig. 2b and Supplementary Fig. 6, scanning electron microscopy (SEM) was conducted to characterize the changes on the morphology of AlGaIn NWs before and after CBD deposition. In the top half of Fig. 2b, the morphology of the pristine AlGaIn specimen shows that the as-grown AlGaIn nanowires by MBE are vertically arranged on substrate and have a uniform morphology in size. After CBD modification, the AlGaIn nanowires are coated by copper-oxides thin layer and the CuO_x layer exhibiting the concavo-convex morphology, which is shown in the bottom half of Fig. 2b. Supplementary Fig. 6 provides the cross-sectional SEM images, where it is obvious that the CuO_x layer resides atop the AlGaIn NWs (~400 nm long)....

18. There are significant issues with English grammar and formatting style, such as extra space, multiple abbreviations, reference format, incorrect spelling and grammatical structure as well as repeated word use.

Response: Thank you for your careful review and valuable feedback regarding the language and formatting issues in our manuscript. We sincerely apologize for these oversights and have tried our best to take the following corrective actions: **Grammar & Spelling:** We have thoroughly proofread the manuscript and corrected all grammatical errors and spelling mistakes. **Formatting & Spacing:** Removed extra spaces and standardized paragraph formatting. Canceled the repeated words use. Ensured consistent abbreviation use.

All the revisions were highlighted in red.